# DOMAIN-FREE ADVERSARIAL SPLITTING FOR DOMAIN GENERALIZATION

## ABSTRACT

Domain generalization is an approach that utilizes several source domains to train the learner to be generalizable to unseen target domain to tackle domain shift issue. It has drawn much attention in machine learning community. This paper aims to learn to generalize well to unseen target domain without relying on the knowledge of the number of source domains and domain labels. We unify adversarial training and meta-learning in a novel proposed Domain-Free Adversarial Splitting (DFAS) framework. In this framework, we model the domain generalization as a learning problem that enforces the learner to be able to generalize well for any train/val subsets splitting of the training dataset. To achieve this goal, we propose a min-max optimization problem which can be solved by an iterative adversarial training process. In each iteration, it adversarially splits the training dataset into train/val subsets to maximize domain shift between them using current learner, and then updates the learner on this splitting to be able to generalize well from train-subset to val-subset using meta-learning approach. Extensive experiments on three benchmark datasets under three different settings on the source and target domains show that our method achieves state-of-the-art results and confirm the effectiveness of our method by ablation study. We also derive a generalization error bound for theoretical understanding of our method.

## 1 INTRODUCTION

Deep learning approach has achieved great success in image recognition (He et al., 2016; Krizhevsky et al., 2012; Simonyan & Zisserman, 2014). However, deep learning methods mostly succeed in the case that the training and test data are sampled from the same distribution (*i.e.*, the *i.i.d.* assumption). However, this assumption is often violated in real-world applications since the equipments/environments that generate data are often different in training and test datasets. When there exists distribution difference (*domain shift (Torralba & Efros, 2011)*) between training and test datasets, the performance of trained model, *i.e., learner*, will significantly degrade.

To tackle the domain shift issue, domain adaptation approach (Pan & Yang, 2010; Daume III & Marcu, 2006; Huang et al., 2007) learns a transferable learner from source domain to target domain. Domain adaptation methods align distributions of different domains either in feature space (Long et al., 2015; Ganin et al., 2016) or in raw pixel space (Hoffman et al., 2018), which relies on unlabeled data from target domain at training time. However, in many applications, it is unrealistic to access the unlabeled target data, therefore this prevents us to use domain adaptation approach in this setting, and motivates the research on the learning problem of domain generalization.

Domain generalization (DG) approach (Blanchard et al., 2011; Muandet et al., 2013) commonly uses several source domains to train a learner that can generalize to an unseen target domain. The underlying assumption is that there exists a latent domain invariant feature space across source domains and unseen target domain. To learn the domain invariant features, (Muandet et al., 2013; Ghifary et al., 2015; Li et al., 2018b) explicitly align distributions of different source domains in feature space. (Balaji et al., 2018; Li et al., 2019b; 2018a; Dou et al., 2019) split source domains into meta-train and meta-test to simulate domain shift and train learner in a meta-learning approach. (Shankar et al., 2018; Carlucci et al., 2019; Zhou et al., 2020; Ryu et al., 2020) augment images or features to train learner to enhance generalization capability.

Conventional domain generalization methods assume that the domain labels are available. But in a more realistic scenario, the domain labels may be unknown (Wang et al., 2019). To handle this domain-free setting, Carlucci et al. (2019) combines supervised learning and self-supervised learning to solve jigsaw puzzles of the training images. Matsuura & Harada (2020) divides samples into several latent domains via clustering and trains a domain invariant feature extractor via adversarial training. Huang et al. (2020) discards the dominant activated features, forcing the learner to activate remaining features that correlate with labels. Another line of works (Volpi et al., 2018; Qiao et al., 2020) tackle the single source setting that the training set comprises a single domain, and the train and test data are from different domains.

In this work, we focus on a general learning scenario of domain generalization as follows. First, we do not know the domain label of each data and do not assume that there are several domains in the training dataset. Second, we do not assume that the training and test data are from different domains (*e.g.*, styles). However, the previous domain-free DG methods (Matsuura & Harada, 2020) commonly evaluate on the datasets (*e.g.*, PACS) composed of several domains though they do not use domain labels in training.

In our domain-free setting, we do not assume and know the domains in the training dataset, we therefore model domain generalization as a learning problem that the learner should be able to generalize well for any splitting of train/val subsets, *i.e.*, synthetic source/target domains, over the training dataset. This explicitly enforces that the trained learner should be generalizable for any possible domain shifts within the training dataset.

To achieve this goal, we propose an adversarial splitting model that is a min-max optimization problem, due to the difficulty of enumerating all splittings. In this min-max problem, we adversarially split training dataset to train/val subsets by maximizing the domain shift between them based on the given learner, and then update learner by minimizing the prediction error on val-subset using meta-learning approach given the splitting. By optimizing this min-max problem, we enforce the learner to generalize well even in the worst-case splitting. We also investigate $L_2$-normalization of features in our domain generalization method. It is surprisingly found that $L_2$-normalization can improve performance of learner and mitigate gradient explosion in the meta-learning process of DG. We further theorectically analyze the underlying reasons for this finding. This proposed domain generalization approach is dubbed *Domain-Free Adversarial Splitting, i.e., DFAS*.

To verify the effectiveness of our method, we conduct extensive experiments on benchmark datasets of PACS, Office-Home and CIFAR-10 under different settings with multiple/single source domains. In experiments that the training data are from several source domains, our method achieves state-of-the-art results on both PACS and Office-Home datasets. We also find that our method significantly outperforms baselines in experiments that the training data are from a single source domain on PACS and CIFAR-10. We also confirm the effectiveness of our method by ablation study.

Based on domain adaptation theory, we also derive an upper bound of the generalization error on unseen target domain. We analyze that the terms in this upper bound are implicitly minimized by our method. This theoretical analysis partially explains the success of our method.

## 2 RELATED WORKS

We summarize and compare with related domain generalization (DG) methods in two perspectives, *i.e.*, DG with domain labels and DG without domain labels.

**DG with domain labels.** When the domain labels are available, there are three categories of methods for DG. First, (Muandet et al., 2013; Ghifary et al., 2015; Li et al., 2018b; Piratla et al., 2020) learn domain invariant features by aligning feature distributions or by common/specific feature decomposition. Second, (Li et al., 2019a; Balaji et al., 2018; Li et al., 2019b; 2018a; Dou et al., 2019; Du et al., 2020a;b) are based on meta-learning approach that splits given source domains into meta-train and meta-test domains and trains learner in an episodic training paradigm. Third, (Shankar et al., 2018; Carlucci et al., 2019; Zhou et al., 2020; Wang et al., 2020) augment fake domain data to train learner for enhancing generalization capability of learner. Our method may mostly relate to the above second category of methods. But differently, we consider the DG problem in domain-free setting and adversarially split training dataset to synthesize domain shift in a principled min-max optimization method, instead of using leave-one-domain-out splitting in these methods.

**DG without domain labels.** When the domain label is unavailable, to enhance generalization ability of learner, Wang et al. (2019) extracts robust feature representation by projecting out superficial patterns like color and texture. Carlucci et al. (2019) proposes to solve jigsaw puzzles of the training images. Matsuura & Harada (2020) divides samples into several latent domains via clustering and learns domain invariant features via adversarial training of feature extractor and domain discriminator. Huang et al. (2020) discards the dominant activated features, forcing the learner to activate remaining features that correlate with labels. Volpi et al. (2018) and Qiao et al. (2020) propose adversarial data augmentation to tackle the setting that the training set comprises a single domain. In methodology, these methods either explicitly force the learner to extract robust features (Wang et al., 2019; Matsuura & Harada, 2020; Huang et al., 2020) or augment new data to increase training data (Carlucci et al., 2019; Qiao et al., 2020; Volpi et al., 2018). While our method is a novel meta-learning approach for DG by introducing adversarial splitting of training dataset during training, without relying on data/domain augmentation.

## 3 METHOD

In our setting, since we do not assume and know the domains in the training dataset, the training data could be independently sampled from several underlying source domains or just from a single source domain. We denote $S = \{(x_i, y_i)\}_{i=1}^{N}$ as the training dataset. Our goal is to train the learner with $S$ that can generalize well on an unseen target domain.

In the following sections, we introduce details of our proposed model in Sect. 3.1, followed by its optimization method in Sect. 3.2. We also investigate $L_2$-normalization for domain generalization in Sect. 3.3. Theoretical analysis for our method is presented in Sect. 4. Experimental results are reported in Sect. 5. Sect. 6 concludes this paper.

### 3.1 DOMAIN-FREE ADVERSARIAL SPLITTING MODEL

As mentioned in Sect. 1, we model DG as a learning problem that enforces the learner to be able to generalize well for any train/val subsets splitting of the training dataset. The learner is trained using meta-learning approach (Finn et al., 2017). To formulate our idea mathematically, we first introduce some notations. We denote $f$ as a function/learner ($f$ could be a deep neural network, *e.g.*, ResNet (He et al., 2016)) that outputs classification score of the input image, $l$ as the loss such as cross-entropy, $S_t$ and $S_v$ as the train-subset and val-subset respectively such that $S = S_t \cup S_v$ and $S_t \cap S_v = \emptyset$. The formulated optimization problem for domain generalization is

$$\min_{w} \frac{1}{|\Gamma_\xi|} \sum_{S_v \in \Gamma_\xi} \mathcal{L}(\theta(w); S_v) + \mathcal{R}(w)$$
$$s.t. \ \ \theta(w) = \arg\min_{\theta} \mathcal{L}(\theta; S_t, w), \ \ S_t = S - S_v. \tag{1}$$

In Eq. (1), $\Gamma_\xi = \{S_v : S_v \subset S, |S_v| = \xi\}$ is the set of all possible val-subsets of $S$ with length of $\xi$, $S_t = S - S_v$ is train-subset paired with each $S_v$, $\mathcal{L}(\theta(w); S_v) = \frac{1}{|S_v|} \sum_{(x,y) \in S_v} l(f_{\theta(w)}(x), y)$ is the loss on $S_v$, where $\theta(w)$ is the parameters of $f$, $\mathcal{L}(\theta; S_t, w)$ is $\mathcal{L}(\theta; S_t)$ with $\theta$ initialized by $w$ and $\mathcal{R}(w)$ is regularization term. In the optimization model of Eq. (1), the parameter $\theta(w)$ of learner trained on $S_t$ is treated as a function of the initial parameter $w$. To force the learner trained on $S_t$ to generalize well on $S_v$, we directly minimize the loss $\mathcal{L}(\theta(w); S_v)$, dubbed *generalization loss*, on val-subset $S_v$, *w.r.t.* the parameter $\theta(w)$ trained on $S_t$. Solving Eq. (1) will force the learner to be able to generalize well from any train-subset to corresponding val-subset.

Since $|\Gamma_\xi|$ may be extremely large, it is infeasible to enumerate all possible train/val splittings. Thus, we propose the following adversarial splitting model instead,

$$\min_{w} \max_{S_v \in \Gamma_\xi} \mathcal{L}(\theta(w); S_v) + \mathcal{R}(w)$$
$$s.t. \ \ \theta(w) = \arg\min_{\theta} \mathcal{L}(\theta; S_t, w), \ \ S_t = S - S_v. \tag{2}$$

In the min-max problem of Eq. (2), the train/val ($S_t/S_v$) splitting is optimized to maximize the generalization loss to increase the domain shift between train and val subsets by finding the hardest splitting to the learner. While $w$ is optimized by minimizing the generalization loss of learner over

the splitting. Solving this adversarial splitting optimization model in Eq. (2) enforces the learner to be generalizable even for the worst-case splitting. We therefore expect that the trained learner is robust to the domain shifts within the training dataset. For the regularization term $\mathcal{R}(w)$, we set it to be the training loss on $S_t$ (*i.e.*, $\mathcal{R}(w) = \mathcal{L}(w; S_t)$), which additionally constrains that the learner with parameter $w$ should be effective on $S_t$ (Li et al., 2018a). The effect of the hyper-parameter $\xi$ will be discussed in Appendix A.2.

In conventional adversarial machine learning, adversarial training is imposed on adversarial samples and learner to increase robustness of the learner to adversarial corruption (Goodfellow et al., 2015). While in our optimization model of Eq. (2), adversarial training is conducted on data splitting and learner to force the learner to be robust to domain shift between train/val subsets. Our model bridges adversarial training and meta-learning. It is a general learning framework for domain generalization and is a complement to adversarial machine learning.

## 3.2 OPTIMIZATION

This section focuses on the optimization of Eq. (2). Since Eq. (2) is a min-max optimization problem, we alternately update $S_v$ and $w$ by fixing the other one as known. We should also consider the inner loop for optimization of $\theta(w)$ in the bi-layer optimization problem of Eq. (2). We next discuss these updating steps in details. The convergence and computational cost of this algorithm will be also discussed in Appendix A.3 and A.4 respectively.

**Inner loop for optimization of $\theta(w)$.** We adopt finite steps of gradient descent to approximate the minimizer $\theta(w)$ of the inner objective $\mathcal{L}(\theta; S_t)$ with initial value $w$. This approximation technique has been introduced in machine learning community several years ago (Sun & Tappen, 2011; Finn et al., 2017; Fan et al., 2018). For convenience of computation, following (Li et al., 2018a; Dou et al., 2019), we only conduct gradient descent by one step as

$$\theta(w) = w - \alpha \nabla_\theta \mathcal{L}(\theta; S_t)|_{\theta=w}, \tag{3}$$

where $\alpha$ is the step size of inner optimization and its effect will be discussed in Appendix A.2.

**Optimizing $w$ with fixed $S_v$.** For convenience, we denote $g_w^t = \nabla_\theta \mathcal{L}(\theta; S_t)|_{\theta=w}$. Fixing $S_v$ ($S_t$ is then fixed), $w$ can also be updated by gradient descent, *i.e.*,

$$w = w - \eta \nabla_w \left( \mathcal{L}(w - \alpha g_w^t; S_v) + \mathcal{R}(w) \right), \tag{4}$$

where $\eta$ is the step size of outer optimization.

**Finding the hardest splitting $S_v$ with fixed $w$.** Fixing $w$, to find $S_v \in \Gamma_\xi$ to maximize $\mathcal{L}(w - \alpha g_w^t; S_v)$, we do first order Taylor expansion for $\mathcal{L}(w - \alpha g_w^t; S_v)$ by $\mathcal{L}(w - \alpha g_w^t; S_v) \approx \mathcal{L}(w; S_v) - \alpha \langle g_w^t, g_w^v \rangle$, where $g_w^v = \nabla_\theta \mathcal{L}(\theta; S_v)|_{\theta=w}$ and $\langle \cdot, \cdot \rangle$ denotes the inner product. From the definition of $\mathcal{L}, g_w^t$ and $g_w^v$, the optimization problem of $\max_{S_v \in \Gamma_\xi} \{ \mathcal{L}(w; S_v) - \alpha \langle g_w^t, g_w^v \rangle \}$ can be written as $\max_{S_v \in \Gamma_\xi} \{ \frac{1}{|S_v|} \sum_{(x,y) \in S_v} l\left(f_w(x), y\right) - \alpha \langle \nabla_w l(f_w(x), y), g_w^t \rangle \}$. This problem is equivalent to the following splitting formulation:

$$\max_{S_v, A} \sum_{(x,y) \in S_v} l\left(f_w(x), y\right) - \alpha \langle \nabla_w l(f_w(x), y), A \rangle \quad s.t. \quad A = g_w^t, S_v \in \Gamma_\xi, \tag{5}$$

where we introduced an auxiliary variable $A$. Eq. (5) can be solved by alternatively updating $S_v$ and $A$. Given $A$, we compute and rank the values of $l\left(f_w(x), y\right) - \alpha \langle \nabla_w l(f_w(x), y), A \rangle$ for all $(x, y) \in S$ and select the largest $\xi$ samples to constitute the $S_v$. Given $S_v$ ($S_t$ is then given), we update $A$ by $A = g_w^t = \frac{1}{|S_t|} \sum_{(x,y) \in S_t} \nabla_w l(f_w(x), y)$. We also discuss details and convergence of this alternative iteration in Appendix C. Since computing gradient *w.r.t.* all parameters is time and memory consuming, we only compute gradient *w.r.t.* parameters of the final layer of learner $f$.

## 3.3 $L_2$-NORMALIZATION FOR EXTRACTED FEATURE

$L_2$-normalization has been used in face recognition (Liu et al., 2017; Wang et al., 2018) and domain adaptation (Saito et al., 2019; Gu et al., 2020), but is rarely investigated in domain generalization. We investigate $L_2$-normalization in domain generalization in this paper. It is found surprisingly in experiments that $L_2$-normalization not only improves the performance of learner (see Sect. 5.4), but

also mitigates gradient explosion (see Sect. 5.5) that occurs frequently during the training of meta-learning for DG (Finn et al., 2017; Dou et al., 2019). We next discuss details of $L_2$-normalization in our method and analyze why $L_2$-normalization mitigates gradient explosion.

**Feature $L_2$-normalization.** The $L_2$-normalization is used as a component of our learner $f$. Specifically, we decompose $f$ into feature extractor $f^e$ (*e.g.*, the convolutional layers of ResNet), the transform $f^n$ representing $L_2$-normalization and classifier $f^c$, *i.e.*, $f = f^c \circ f^n \circ f^e$. The feature of input image $x$ extracted by $f^e$ is fed to $f^n$ to output an unit vector $z$, *i.e.*, $z = f^n(f^e(x)) = \frac{f^e(x)}{\|f^e(x)\|}$. The classifier $f^c$ consists of unit weight vectors $W = [w_1, w_2, \cdots, w_K]$, where $K$ is the number of classes and $\|w_k\| = 1, \forall k$. $f^c$ takes $z$ as an input and outputs the classification score vector $\sigma_{m,s}(W^T z)$. $\sigma_{m,s}(\cdot)$ is the marginal softmax function defined by

$$[\sigma_{m,s}(W^T z)]_k = \frac{\exp(s(w_k^T z - m\mathbb{I}_{\{k=y\}}))}{\sum_{k'=1}^{K} \exp(s(w_{k'}^T z - m\mathbb{I}_{\{k'=y\}}))}, \quad k = 1, 2, \cdots, K, \tag{6}$$

where $y$ is the label of $x$, $[\cdot]_k$ indicates the $k$-th element, $\mathbb{I}_{\{a\}}$ is the indicator function that returns 1 if $a$ is true, 0 otherwise, $m$ and $s$ are hyper-parameters indicating margin and radius respectively.

**Analysis of mitigating gradient explosion.** We find that $L_2$-normalization mitigates gradient explosion in the training of meta-learning for domain generalization. For the sake of simplicity, we analyze gradient norm of loss *w.r.t.* parameters of $f^c$ in the meta-learning process of domain generalization, with $f^e$ as fixed function. Without loss of generality, we consider the case that $K = 2$ (*i.e.*, binary classification), $s = 1$ and $m = 0$. In this case, we have the following proposition.

**Proposition 1.** *Under the above setting, if the input feature of $f^c$ is $L_2$-normalized, the gradient norm of loss w.r.t. parameters of $f^c$ in the meta-learning process of DG is bounded.*

*Sketch of proof.* Given feature $z$, the loss of binary classification is $\mathcal{L}(w; z) = -y \log(\sigma(w^T z)) - (1 - y)\log(1 - \sigma(w^T z))$, where $\sigma$ is the sigmoid function. Let $w' = w - \alpha \nabla_w \mathcal{L}(w; z)$, then $\nabla_w \mathcal{L}(w'; z) = (I - \alpha H)\nabla_{w'} \mathcal{L}(w'; z)$, where $H$ is the Hessian matrix. The gradient norm $\|\nabla_w \mathcal{L}(w'; z)\| \leq \|I - \alpha H\| \|\nabla_{w'} \mathcal{L}(w'; z)\| \leq (1 + |\alpha| \|H\|) \|\nabla_{w'} \mathcal{L}(w'; z)\|$. Since $\nabla_{w'} \mathcal{L}(w'; z) = (p - y)z$ and $H = p(1 - p)zz^T$ where $p = \sigma(w^T z)$, $\|H\| = \sup_{u:\|u\|=1} \|Hu\| \leq \sup_{u:\|u\|=1} \|zz^T u\| \leq \|z\|^2$ and $\|\nabla_{w'} \mathcal{L}(w'; z)\| \leq \|z\|$. If $\|z\| = 1$, we have $\|\nabla_w \mathcal{L}(w'; z)\| \leq 1 + |\alpha|$.

According to Proposition 1, $L_2$-normalization can mitigate gradient explosion under the above setting. The analysis of gradient norm of loss *w.r.t.* parameters of both $f^c$ and $f^e$ in the meta-learning process is much more complex, left for our future work.

## 4 THEORETICAL ANALYSIS

This section presents theoretical understanding of our method. We first derive a generalization error bound on target domain in theorem 1 for the general setting of meta-learning for DG. Then, based on theorem 1, we theoretically explain the reason on the success of our method.

Without loss of generality, we consider binary classification problem. We denote $\mathcal{H}$ as the set of all possible $f$, *i.e.*, $\mathcal{H} = \{f_w : w \in R^M\}$, where $M$ is the number of parameters. For any $S_v \in \Gamma_\xi$ and $S_t = S - S_v$, we let $\mathcal{H}_{S_t} = \{f_{\theta(w)} : \theta(w) = \arg\min_\theta \mathcal{L}(\theta; S_t, w), w \in R^M\}$. The meta-learning approach for DG is to find a function in $\mathcal{H}_{S_t}$ to minimize classification loss on $S_v$. Note that, although the training samples in $S$ may be sampled from several distributions, they can still be seen as being *i.i.d.* sampled from a mixture of these distributions. We next respectively denote $\mathcal{P} = \sum_{d=1}^{D} \beta_d \mathcal{P}_d$ as the mixture distribution with $\beta_d$ representing the sampling ratio of the $d$-th source domain, $\epsilon_\mathcal{Q}^\Psi(f) = \mathbb{E}_{(x,y)\sim\mathcal{Q}}[\mathbb{I}_{\{\Psi(f(x))\neq y\}}]$ as the generalization error on distribution $\mathcal{Q}$ of unseen target domain, $\hat{\epsilon}_{S_v}^\Psi(f) = \frac{1}{|S_v|} \sum_{(x,y)\in S_v} \mathbb{I}_{\{\Psi(f(x))\neq y\}}$ as the empirical error on $S_v$, $VC(\mathcal{H})$ as the VC-dimension of $\mathcal{H}$, and $\Psi(\cdot)$ as the prediction rule such as the Bayes Optimal Predictor. Based on the domain adaptation theory (Ben-David et al., 2007; 2010) and inspired by the analysis in (Saito et al., 2019), we have the following theorem.

**Theorem 1.** *Let $\gamma$ be a constant, assume $\mathbb{E}_\mathcal{Q}[\mathbb{I}_{\{l(f(x),y)>\gamma\}}] \geq \mathbb{E}_\mathcal{P}[\mathbb{I}_{\{l(f(x),y)>\gamma\}}]$, then given any $S_v \in \Gamma_\xi$ and $S_t = S - S_v$ and for any $\delta \in (0, 1)$, with probability at least $1 - 2\delta$, we have $\forall f \in \mathcal{H}_{S_t}$,*

Table 1: Results of *MSDS* experiment on PACS based on ResNet18 and ResNet50.

| Backbone | Target | D-SAM | JiGen | MASF | MMLD | MetaReg | RSC | Base-line | DFAS (ours) |
|---|---|---|---|---|---|---|---|---|---|
| ResNet18 | A | 77.3 | 79.4 | 80.3 | 81.3 | 83.7 | 83.4 | $80.2^{\pm0.4}$ | $\mathbf{84.2}^{\pm0.1}$ |
| | C | 72.4 | 75.3 | 77.2 | 77.2 | 77.2 | **80.3** | $75.5^{\pm0.5}$ | $79.5^{\pm0.3}$ |
| | P | 95.3 | 96.0 | 95.0 | **96.1** | 95.5 | 96.0 | $95.9^{\pm0.1}$ | $95.8^{\pm0.1}$ |
| | S | 77.8 | 71.4 | 71.7 | 72.3 | 70.3 | 80.9 | $70.1^{\pm0.9}$ | $\mathbf{82.1}^{\pm0.4}$ |
| | Avg | 80.7 | 80.5 | 81.0 | 81.8 | 81.7 | 85.1 | 80.4 | **85.4** |
| ResNet50 | A | - | - | 82.9 | - | 87.2 | 87.9 | $86.1^{\pm0.2}$ | $\mathbf{89.1}^{\pm0.1}$ |
| | C | - | - | 80.5 | - | 79.2 | 82.2 | $79.2^{\pm0.4}$ | $\mathbf{84.6}^{\pm0.2}$ |
| | P | - | - | 95.0 | - | 97.6 | **97.9** | $97.6^{\pm0.1}$ | $96.8^{\pm0.2}$ |
| | S | - | - | 72.3 | - | 70.3 | 83.5 | $70.3^{\pm0.7}$ | $\mathbf{85.6}^{\pm0.3}$ |
| | Avg | - | - | 82.7 | - | 83.6 | 87.8 | 83.3 | **89.0** |

$$\epsilon_{\mathcal{Q}}^{\Psi_l}(f) \leq \hat{\epsilon}_{S_v}^{\Psi_l}(f) + B(S_v) + 2\sqrt{\frac{8}{\xi}\left(C_2 + \frac{4}{\delta}\right)} + C_3, \tag{7}$$

*where*

$$B(S_v) = C_1 - \inf_{f' \in \mathcal{H}_{S_t}} \frac{1}{|S_v|} \sum_{(x,y) \in S_v} \mathbb{I}_{\{l(f'(x),y) > \gamma\}}, \tag{8}$$

$C_1 = \sup_{S_v' \in \Gamma_\xi} \sup_{f' \in \mathcal{H}_{S-S_v'}} \mathbb{E}_{\mathcal{Q}}[\mathbb{I}_{\{l(f'(x),y)>\gamma\}}]$, $C_2 = \sup_{S_v' \in \Gamma_\xi} VC(\mathcal{H}_{S-S_v'}^{\Psi_l}) \log \frac{2e\xi}{VC(\mathcal{H}_{S-S_v'}^{\Psi_l})}$,
$C_3 \geq \sup_{S_v' \in \Gamma_\xi} \inf_{f' \in \mathcal{H}_{S-S_v'}} \{\epsilon_{\mathcal{P}}^{\Psi_l}(f') + \epsilon_{\mathcal{Q}}^{\Psi_l}(f')\}$, $\mathcal{H}_{S-S_t}^{\Psi_l} = \{\Psi_l \circ f : f \in \mathcal{H}_{S-S_t}\}$, $\Psi_l$ *is a loss-related indicator defined by*

$$\Psi_l(f(x)) = \begin{cases} 1 & if\, l(f(x),y) > \gamma. \\ 0 & otherwise\,. \end{cases} \tag{9}$$

Proof is given in Appendix D. The assumption of $\mathbb{E}_{\mathcal{Q}}[\mathbb{I}_{\{l(f(x),y)>\gamma\}}] \geq \mathbb{E}_{\mathcal{P}}[\mathbb{I}_{\{l(f(x),y)>\gamma\}}]$ in theorem 1 is realistic because the data of $\mathcal{Q}$ is not accessed at training time, and the learner trained on data of $\mathcal{P}$ should have smaller classification loss on $\mathcal{P}$ than $\mathcal{Q}$. In theorem 1, $C_1, C_2, C_3$ are constants to $f$. In Eq. (7), the generalization error $\epsilon_{\mathcal{Q}}^{\Psi_l}(f)$ on $\mathcal{Q}$ can be bounded by the empirical error $\hat{\epsilon}_{S_v}^{\Psi_l}(f)$ on $S_v$, the term $B(S_v)$ that measures the discrepancy between $\mathcal{P}$ and $\mathcal{Q}$, and the last two constant terms in Eq. (7).

To obtain lower $\epsilon_{\mathcal{Q}}^{\Psi_l}(f)$, we need to minimize $\hat{\epsilon}_{S_v}^{\Psi_l}(f)$ and $B(S_v)$. Minimizing $B(S_v)$ *w.r.t.* $S_v$ is equivalent to

$$\max_{S_v \in \Gamma_\xi} \inf_{f \in \mathcal{H}_{S_t}} \frac{1}{|S_v|} \sum_{(x,y) \in S_v} \mathbb{I}_{\{l(f(x),y) > \gamma\}}. \tag{10}$$

Intuitively, Eq. (10) means to find a $S_v \in \Gamma_\xi$ such that the infimum ratio of examples in $S_v$ having loss greater than $\gamma$ is maximized. This min-max problem of Eq. (10) for computing the error bound bears the similar idea as our min-max problem. Our adversarial splitting model in Eq. (2) can implicitly realize the goal of Eq. (10) and meanwhile ensure lower $\hat{\epsilon}_{S_v}^{\Psi_l}(f)$ for any $S_v$.

The maximization in Eq. (10) corresponds to our adversarial splitting that finds the hardest val-subset $S_v$ for the learner in Eq. (2). The infimum in Eq. (10) corresponds to the minimization of the loss in Eq. (2) on $S_v$ *w.r.t.* the learner parameterized by $\theta(w)$. Instead of using indicator function $\mathbb{I}$ in Eq. (10), in our model of Eq. (2), we choose differentiable classification loss for easier optimization.

## 5 EXPERIMENTS

We verify the effectiveness of our method in three types of experimental settings: *Multi Source with Domain Shift (MSDS)* that the training data are from several source domains and there exists domain

Table 2: Results of *MSDS* experiment on Office-Home based on ResNet18 and ResNet50.

| Backbone | Target | D-SAM | JiGen | RSC | Base-line | DFAS (ours) | Backbone | Target | Base-line | DFAS (ours) |
|---|---|---|---|---|---|---|---|---|---|---|
| ResNet18 | Ar | 58.0 | 53.0 | 58.4 | $56.7^{\pm0.3}$ | $\mathbf{62.0}^{\pm0.2}$ | ResNet50 | Ar | $64.9^{\pm0.4}$ | $\mathbf{70.2}^{\pm0.2}$ |
| | Cl | 44.4 | 47.5 | 47.9 | $47.6^{\pm0.2}$ | $\mathbf{48.6}^{\pm0.3}$ | | Cl | $51.8^{\pm0.2}$ | $\mathbf{53.5}^{\pm0.4}$ |
| | Pr | 69.2 | 71.5 | **71.6** | $71.4^{\pm0.1}$ | $71.4^{\pm0.1}$ | | Pr | $76.5^{\pm0.2}$ | $\mathbf{77.4}^{\pm0.1}$ |
| | Rw | 71.5 | 72.8 | 74.5 | $72.9^{\pm0.3}$ | $\mathbf{75.2}^{\pm0.1}$ | | Rw | $79.4^{\pm0.3}$ | $\mathbf{80.2}^{\pm0.2}$ |
| | Avg | 60.8 | 61.2 | 63.1 | 62.2 | **64.3** | | Avg | 68.2 | **70.3** |

Table 3: Results of *SSDS* experiment on PACS based on ResNet18.

| Method | A→C | A→P | A→S | C→A | C→P | C→S | P→A | P→C | P→S | S→A | S→C | S→P | Avg |
|---|---|---|---|---|---|---|---|---|---|---|---|---|---|
| JiGen | 57.0 | **96.1** | 50.0 | 65.3 | 85.5 | 65.9 | 62.4 | 27.2 | 35.5 | 26.6 | 41.1 | 42.8 | 54.6 |
| SagNet | 67.1 | 95.7 | 56.8 | **72.1** | 85.7 | 69.2 | 69.8 | 35.1 | 40.7 | 41.1 | 62.9 | 46.2 | 61.9 |
| BaseLine | $63.7^{\pm.3}$ | $95.6^{\pm.1}$ | $63.5^{\pm.4}$ | $72.0^{\pm.3}$ | $\mathbf{86.5}^{\pm.1}$ | $73.3^{\pm.2}$ | $68.4^{\pm.4}$ | $32.7^{\pm.5}$ | $42.2^{\pm.4}$ | $41.6^{\pm.5}$ | $60.3^{\pm.2}$ | $49.3^{\pm.3}$ | 62.4 |
| DFAS (ours) | $\mathbf{67.5}^{\pm.2}$ | $94.5^{\pm.1}$ | $\mathbf{67.1}^{\pm.3}$ | $69.0^{\pm.2}$ | $\mathbf{86.5}^{\pm.1}$ | $\mathbf{73.8}^{\pm.3}$ | $\mathbf{70.2}^{\pm.2}$ | $\mathbf{36.1}^{\pm.4}$ | $\mathbf{52.1}^{\pm.2}$ | $\mathbf{56.7}^{\pm.2}$ | $\mathbf{67.9}^{\pm.3}$ | $\mathbf{57.4}^{\pm.1}$ | **66.6** |

shift between training and test data, *Single Source with Domain Shift (SSDS)* that the training data are from a single source domain and there exists domain shift between training and test data, and *Same Source and Target Domain (SSTD)* that the training and test data are from a same single domain. The source codes will be released online.

We conduct experiments on three benchmark datasets. **PACS** (Li et al., 2017) contains four domains, including art painting (A), cartoon (C), photo (P), sketch (S), sharing seven classes. **Office-Home** (Volpi et al., 2018), a dataset widely used in domain adaptation and recently utilized in domain generalization, consists of four domains: Art (Ar), Clipart (Cl), Product (Pr), Real World (Rw), sharing 65 classes. Both of these two datasets are utilized to conduct experiments in settings of *MSDS* and *SSDS*. **CIFAR-10** (Krizhevsky et al., 2009) is taken for the experimental setting of *SSTD*.

## 5.1 Type I: Multi Source with Domain Shift (MSDS)

In the setting of *MSDS*, following (Carlucci et al., 2019), we use leave-one-domain-out cross-validation, *i.e.*, training on three domains and testing on the remaining unseen domain, on PACS and Office-Home. Note that the domain labels are not used during training. We adopt ResNet18 and ResNet50 (He et al., 2016) pre-trained on ImageNet (Russakovsky et al., 2015). For each of them, the last fully-connected layer is replaced by a bottleneck layer, then the corresponding network is taken as feature extractor $f^e$. Full implementation details are reported in Appendix B.

We compare our method with several state-of-the-art methods, including D-SAM (D'Innocente & Caputo, 2018), JiGen (Carlucci et al., 2019), MASF (Dou et al., 2019), MMLD (Matsuura & Harada, 2020), MetaReg (Balaji et al., 2018), RSC (Huang et al., 2020). The results on PACS and Office-Home are reported in Table 1 and Table 2 respectively. Our DFAS achieves state-of-the-art results based on both ResNet18 and ResNet50 on both PACS (85.4%, 89.0%) and Office-Home (64.3%, 70.3%), outperforming RSC by 0.3% and 1.2% on PACS based on ResNet18 and ResNet50 respectively, and by 1.2% on Office-Home based on ResNet18 (these methods do not conduct experiment on Office-Home using ResNet50). Compared with Baseline that directly aggregates three source domains to train learner with standard fully-connected layer as classifier $f^c$, our method of DFAS improves its performance by 5.0% and 5.7% on PACS based on ResNet18 and ResNet50 respectively, and by 2.1% and 2.1% on Office-Home based on ResNet18 and ResNet50 respectively. In Table 1, on PACS, DFAS significantly outperforms Baseline in almost all tasks except when P is taken as target domain. Note that, in the task that domain S, of which the style is extremely different from rest three domains, is target domain, our DFAS boosts the accuracy of Baseline by 12.0% and 15.3% based on ResNet18 and ResNet50 respectively. This indicates that our method can generalize well when domain shift is large. Office-Home is challenging for DG since the number of classes is larger than other datasets. As shown in Table 2, our DFAS outperforms Baseline stably in almost all tasks on Office-Home. These performance improvements demonstrate the effectiveness of our method in the case that training data are from multi-source domains and the unseen target domain is different from source domains.

## 5.2 TYPE II: SINGLE SOURCE WITH DOMAIN SHIFT (SSDS)

We conduct this type of experiment on PACS based on ResNet18. In this experiment, we train learner on one domain and test on each of the rest three domains, resulting in total 12 tasks. Implementation details are shown in Appendix B. Our method is compared with related methods, including Baseline that directly trains learner with standard fully-connected layer as classifier $f^c$ on the source domain, Jien (Carlucci et al., 2019) and SagNet (Nam et al., 2019). Results are reported in Table 3. Our method of DFAS outperforms Baseline and SagNet by 4.2% and 4.7% respectively. We observe that our method outperforms Baseline in 10 tasks among all 12 tasks. The performance boosts are large in tasks when domain S is set to be source domain. These performance improvements demonstrate the effectiveness of our method in the case that training data are from single source domain and the unseen target domain is different from source domain.

Table 4: Results of *SSTD* experiment on CIFAR-10 based on ResNet18.

| Training Size | 100 | 600 | 1000 | 3000 | 5000 | 10000 | Total | Avg |
|---|---|---|---|---|---|---|---|---|
| JiGen | 38.5 | 62.9 | 66.1 | 76.5 | 79.2 | 83.9 | 91.2 | 71.2 |
| MMLD | 40.5 | 59.8 | 65.1 | 75.8 | 78.8 | 82.5 | 90.7 | 70.5 |
| Baseline | $39.3^{\pm.4}$ | $62.3^{\pm.3}$ | $66.4^{\pm.6}$ | $76.3^{\pm.3}$ | $82.0^{\pm.4}$ | $85.8^{\pm.5}$ | $94.9^{\pm.1}$ | 72.4 |
| DFAS (ours) | $\mathbf{43.9}^{\pm.5}$ | $\mathbf{64.3}^{\pm.2}$ | $\mathbf{69.4}^{\pm.3}$ | $\mathbf{80.0}^{\pm.4}$ | $\mathbf{83.9}^{\pm.4}$ | $\mathbf{87.2}^{\pm.3}$ | $\mathbf{95.1}^{\pm.0}$ | $\mathbf{74.8}$ |

## 5.3 TYPE III: SAME SOURCE AND TARGET DOMAIN (SSTD)

We also apply our DG method to the common recognition task that the training and test data are from a same domain, *i.e.*, *SSTD*, on CIFAR-10 dataset. To investigate the effect of training size, we sample different sizes of training data from the provided training set (*i.e.*, source domain). Implementation details are in Appendix B. As shown in Table 4, our DFAS outperforms Baseline, JiGen and MMLD by 2.4%, 3.6% and 4.3% respectively on average. The results of JiGen and MMLD are obtained by running their codes on CIFAR-10. We observe that DFAS outperforms Baseline and compared methods in all different numbers of training data. In general, the performance boost is larger when the number of training data is smaller. This may be because the learner is more possible to be overfitting when the training size is smaller and our DFAS is designed to extract better generalizable features.

## 5.4 ABLATION STUDY

To further verify the effectiveness of each component of our method, we conduct additional ablation experiments on PACS dataset based on ResNet18 in both *MSDS* and *SSDS* setting. The results are reported in Table 5 and Table 6.

Table 5: Additional ablation results on PACS in *MSDS* setting.

| $L_2$-norm | Rand-split | Adv-split | A | C | P | S | Avg |
|---|---|---|---|---|---|---|---|
| ✗ | ✗ | ✗ | 80.2 | 75.5 | **95.9** | 70.1 | 80.4 |
| ✓ | ✗ | ✗ | 82.5 | 77.5 | 95.2 | 75.6 | 82.7 |
| ✗ | ✓ | ✗ | 80.6 | 76.8 | 95.6 | 78.1 | 82.8 |
| ✗ | ✗ | ✓ | 83.1 | 77.0 | 94.8 | 79.3 | 83.6 |
| ✓ | ✓ | ✗ | 83.2 | 78.6 | 95.8 | 79.5 | 84.3 |
| ✓ | ✗ | ✓ | **84.2** | **79.5** | 95.8 | **82.1** | **85.4** |

In Table 5 and 6, $L_2$-norm denotes feature $L_2$-normalization defined in Sect.3.3. Rand-split means the random splitting strategy that we randomly split the train/val subsets at each step of updating the parameters of learner. Adv-split denotes the adversarial splitting model that we update the worst-case splitting by solving the maximization problem in Eq. (5) per epoch in training process.

**Effectiveness of $L_2$-normalization.** In Table 5, $L_2$-norm (82.7%) outperforms Baseline (80.4%) by 2.3% and $L_2$-norm + Adv-split (i.e., DFAS) (85.4%) outperforms Adv-split (83.6%) by 1.8% in *MSDS* setting. In Table 6, $L_2$-norm (63.1%) outperforms Baseline (62.4%) by 0.7% and $L_2$-norm + Adv-split (i.e., DFAS) (66.6%) outperforms Adv-split (65.1%) by 1.5% in *SSDS* setting.

Table 6: Additional ablation results on PACS in *SSDS* setting.

| $L_2$-norm | Rand-split | Adv-split | A→C | A→P | A→S | C→A | C→P | C→S | P→A |
|---|---|---|---|---|---|---|---|---|---|
| × | × | × | 63.7 | **95.6** | 63.5 | 72.0 | 86.5 | 73.3 | 68.4 |
| ✓ | × | × | 64.5 | 95.4 | **67.2** | 69.8 | 87.2 | 73.3 | 67.2 |
| × | ✓ | × | 65.8 | 94.8 | 64.4 | 71.2 | 85.2 | 74.2 | 68.7 |
| × | × | ✓ | **67.6** | 95.2 | 66.2 | **76.9** | **88.4** | 73.5 | **70.4** |
| ✓ | ✓ | × | 67.2 | 95.3 | 63.7 | 70.8 | 85.3 | **75.5** | 70.0 |
| ✓ | × | ✓ | 67.5 | 94.5 | 67.1 | 69.0 | 86.5 | 73.8 | 70.2 |
| $L_2$-norm | Rand-split | Adv-split | P→C | P→S | S→A | S→C | S→P | Avg | |
| × | × | × | 32.7 | 42.2 | 41.6 | 60.3 | 49.3 | 62.4 | |
| ✓ | × | × | 31.7 | 38.5 | 44.3 | 67.9 | 49.8 | 63.1 | |
| × | ✓ | × | 32.2 | 39.5 | 52.4 | 64.6 | 56.6 | 64.1 | |
| × | × | ✓ | 30.5 | 35.3 | 52.7 | 67.7 | 57.3 | 65.1 | |
| ✓ | ✓ | × | **37.2** | 45.9 | 53.5 | 64.5 | 55.8 | 65.4 | |
| ✓ | × | ✓ | 36.1 | **52.1** | **56.7** | **67.9** | **57.4** | **66.6** | |

These performance improvements demonstrate that the feature $L_2$-normalization is useful in domain generalization.

**Effectiveness of adversarial splitting model.** In Table 5, Adv-split (83.6%) outperforms Baseline (80.4%) by 3.2% and $L_2$-norm + Adv-split (i.e., DFAS) (85.4%) outperforms $L_2$-norm (82.7%) by 2.7% in *MSDS* setting. In Table 6, Adv-split (65.1%) outperforms Baseline (62.4%) by 2.7% and $L_2$-norm + Adv-split (i.e., DFAS) (66.6%) outperforms $L_2$-norm (63.1%) by 3.5% in *SSDS* setting. These results indicating that our proposed adversarial splitting model is effective.

**Effectiveness of adversarial splitting over random splitting.** In Table 5, $L_2$-norm + Adv-split (85.4%) outperforms $L_2$-norm + Rand-split (84.3%) by 1.1% and Adv-split (83.6%) outperforms Rand-split (82.8%) by 0.8% in *MSDS* setting. In Table 5, $L_2$-norm + Adv-split (66.6%) outperforms $L_2$-norm + Rand-split (65.4%) by 1.2% and Adv-split (65.1%) outperforms Rand-split (64.1%) by 1.0% in *SSDS* setting. These results demonstrate that the adversarial splitting model outperforms the random splitting strategy in different experimental settings.

Due to space limit, we add more ablation experiments in Appendix A.1 to further compare different splittings, including adversarial splitting, domain-label-based splitting and random splitting.

## 5.5 MITIGATING GRADIENT EXPLOSION BY $L_2$-NORMALIZATION.

To show that $L_2$-normalization can mitigate gradient explosion, we conduct the same experiments independently for 50 times respectively with $L_2$-normalization and without $L_2$-normalization. Then we count the numbers of occurrences of gradient explosion that are reported in Table 7. From Table 7, we can observe that $L_2$-normalization can mitigate gradient explosion.

Table 7: The number of occurrences of gradient explosion.

| | w/o $L_2$-normalization | w/ $L_2$-normalization |
|---|---|---|
| Times | 12/50 | 0/50 |

## 6 CONCLUSION

In this paper, we unify adversarial training and meta-learning in a novel proposed Domain-Free Adversarial Splitting (DFAS) framework to tackle the general domain generalization problem. Extensive experiments show the effectiveness of the proposed method. We are interested in deeper theoretical understanding and more applications of our method in the future work.

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

## A  ANALYSIS

### A.1  COMPARISON OF ADVERSARIAL SPLITTING AND DOMAIN-LABEL-BASED SPLITTING

In the section, we compare our adversarial splitting with domain-label-based splitting that is commonly used in meta-learning-based DG methods. Due to variations of style, poses, sub-classes, etc., the internal inconsistency within dataset is complicated. Domain-label partially capture the inconsistency, while cannot cover all possible internal inconsistency. Our adversarial splitting method does not rely on the domain label. It iteratively finds the hardest train/val splitting to the learner to maximize the inconsistency and train the learner to generalize well for the hardest splitting, in an adversarial training way. This strategy more flexibly investigates the possible inconsistency within training dataset, adaptively to the learner, and can potentially enhance the generalization ability of learner.

Table 8: Values of objective function in Eq. (5) of Adv-split and Label-split.

| Learner | $w_1$ | $w_2$ | $w_3$ | $w_4$ |
|---|---|---|---|---|
| Adv-split | 4.15 | 4.28 | 2.61 | 1.28 |
| Label-split | 3.72 | 2.82 | 1.29 | 0.21 |

We first empirically show that the domain-label-based splitting (denoted as Label-split) is not as hard as our adversarial splitting (Adv-split) to the learner in Table 8. In Table 8, we report the values of objective function in Eq. (5) of Adv-split and Label-split by fixing the learner with different network parameters $w_i$ at different epoch (1th, 2th, 5th and 10th) in the training process. Larger value in the table indicates that the splitting is harder to the learner (i.e., network). It can be observed that the domain-label-based splitting (Label-split) is not as hard as Adv-split to learner.

We also conduct experiments on PACS in *MSDS* setting to fairly compare different splittings, including adversarial splitting (Adv-split), domain-label-based splitting (Label-split) and random splitting (Rand-split). The results are reported in Table 9. Table 9 shows that adversarial splitting outperforms random splitting and domain-label-based splitting when training data is from multiple domains.

When the training data are from only a single domain, our adversarial splitting also performs well (as in Table 3). However, domain-label-based splitting cannot be used in this setting, since there is no domain label available.

Table 9: Results of different splittings.

| Target | A | C | P | S | Avg |
|---|---|---|---|---|---|
| Rand-split | 80.6 | 76.8 | **95.6** | 78.1 | 82.8 |
| Label-split | 81.2 | 75.9 | 94.7 | **80.1** | 83.0 |
| Adv-split | **83.1** | **77.0** | 94.8 | 79.3 | **83.6** |

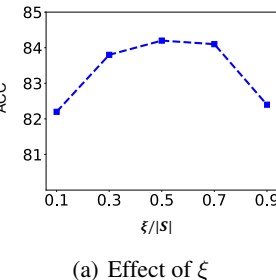

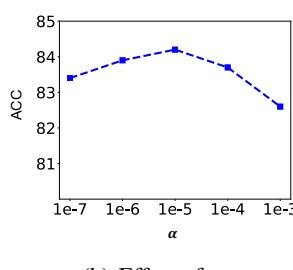

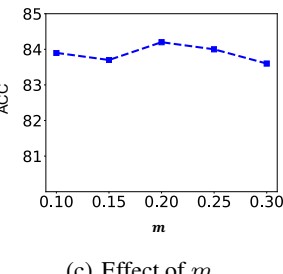

(a) Effect of $\xi$           (b) Effect of $\alpha$           (c) Effect of $m$

Figure 1: Effect of hyper-parameters of $\xi$, $\alpha$ and $m$. We use ResNet18, and domain A is taken as target domain on PACS dataset in the setting of *MSDS*.

## A.2 EFFECT OF HYPER-PARAMETERS

**Effect of hyper-parameter $\xi$.** In Fig. 1(a), we show the performance of our method when varying the hyper-parameter $\xi$, *i.e.,* length of the val-subset $S_v$ in adversarial splitting of training dataset. The best result is obtained when $\xi = \frac{|S|}{2}$, and the results are similar when $\frac{\xi}{|S|}$ ranges from 0.3 to 0.7.

**Effect of hyper-parameter $\alpha$.** We evaluate the effect of $\alpha$ in *MSDS* setting on PACS dataset in Fig. 1(b). From Fig. 1(b), the ACC is stable to the values of $\alpha$ in large range of 1e-6 to 1e-4. Small $\alpha$ results in small step-size for parameter updating in meta-learning framework, and limits the benefits from meta-learning and adversarial splitting. Larger $\alpha$ results in larger step-size for gradient descent based network updating, which may fail to decrease the training loss from the optimization perspective.

**Effect of hyper-parameter $m$.** The effect of $m$ is evaluated in *MSDS* setting on PACS dataset in Fig. 1(c). Fig. 1(c) shows that the result is not sensitive to the value of $m$.

## A.3 CONVERGENCE

We testify the convergence of DFAS with errors and losses in different tasks in Fig. 2. In Fig. 2(a) and Fig. 2(b) , we show the classification error curves on target domains (A and Ar respectively) in the setting of *MSDS*. In Fig. 2(c), we show the training loss of DFAS in task A in *MSDS* setting. These training curves indicates that DFAS converges in the training process. We also observe that DFAS has better stability than Baseline, in Fig. 2(a) and Fig. 2(b).

## A.4 COMPUTATIONAL COST OF ADVERSARIAL SPLITTING AND RANDOM SPLITTING

We compare the computational cost of adversarial splitting and random splitting in this section. Since we only update the worst-case splitting per epoch, instead of at each step of updating parameters, the computational cost is only slightly higher than that of random splitting. To show this, we compare the total training times of the adversarial splitting and random spitting in the same number of steps (20000), as in Table 10.

From Table 10, the training time of Adv-split is only 5.6% (0.33/5.90) higher than Rand-split.

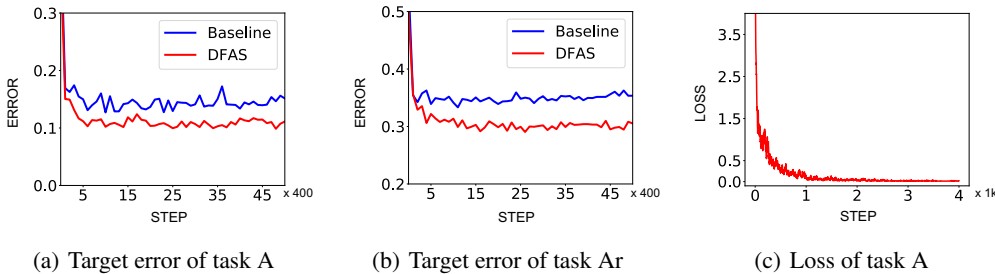

(a) Target error of task A      (b) Target error of task Ar      (c) Loss of task A

Figure 2: Curves of target errors and losses of task A (PACS) and Ar (Office-Home) during training based on ResNet50 in the setting of *MSDS*.

Table 10: Total training time (hour) of the adversarial splitting (Adv-split) and random spitting (Rand-split).

|  | Adv-split | Rand-split |
|---|---|---|
| Time | 6.23h | 5.90h |

## A.5   VISUALIZATION OF FEATURE SPACE

We visualize the feature space learned by our method of DFAS and Baseline (shown in Fig. 3), by t-SNE (Maaten & Hinton, 2008). It appears that DFAS yields better separation of classes and better alignment of distributions of source and unseen target domains, which possibly explains the accuracy improvements achieved by our DFAS.

## B   IMPLEMENTATION DETAILS

For the setting of *MSDS*, we use ResNet18 and ResNet50 (He et al., 2016) pre-trained on ImageNet (Russakovsky et al., 2015). For each of them, the last fully-connected layer is replaced by a bottleneck layer, then the corresponding network is taken as feature extractor $f^e$. The dimension of the bottleneck layer is set to be 512 when the backbone is ResNet18 as in (Saito et al., 2019), and 256 when the backbone is ResNet50 as in (Gu et al., 2020). Following (Gu et al., 2020), $s$ is set to 7.5 for PACS and 10.0 for Office-Home. $m$ is set to 0.2 for PACS and 0.1 for Office-Home. $\xi$ is set to $\frac{|S|}{2}$. SGD with momentum of 0.9 is utilized to update parameters of learner. The learning rate of classifier and bottleneck layer is 10 times of convolutional layers, which is widely adopted in domain adaptation (Long et al., 2015; Ganin et al., 2016). Following (Ganin et al., 2016), the learning rate of convolutional layer is adjusted by $\eta = \frac{0.001}{(1+10p)^{0.75}}$, where $p$ is the optimizing progress linearly changing from 0 to 1. The learning rate $\alpha$ of inner loop optimization is set to $10^{-5}$. The parameters are updated for 20000 steps and the hardest val-subset is updated per 200 steps. The batchsize is set to 64. The running mean and running variance of Batch Normalization (BN) layers are fixed as the pre-trained values on ImageNet during training, which is discussed in (Du et al., 2020a). Due to memory limit, when implementing experiments based on ResNet50, we adopt the first order approximation (Finn et al., 2017) that stops the gradient of $g_w^t$ in Eq. (4) for reducing memory and computational cost.

For the setting of *SSDS*, we conduct experiment based on ResNet18 on PACS. The implementation details are same as *MSDS*. For the setting of *SSTD*, we conduct experiment based on ResNet18 on CIFAR-10. The hyper-parameters of $s$ and $m$ are set to 8.0 and 0.2 respectively. Other implementation details are same as *MSDS* except that, in BN layers, the running mean and running variance are updated.

We implement experiments using Pytorch (Paszke et al., 2019) on a single NVIDIA Tesla P100 GPU.

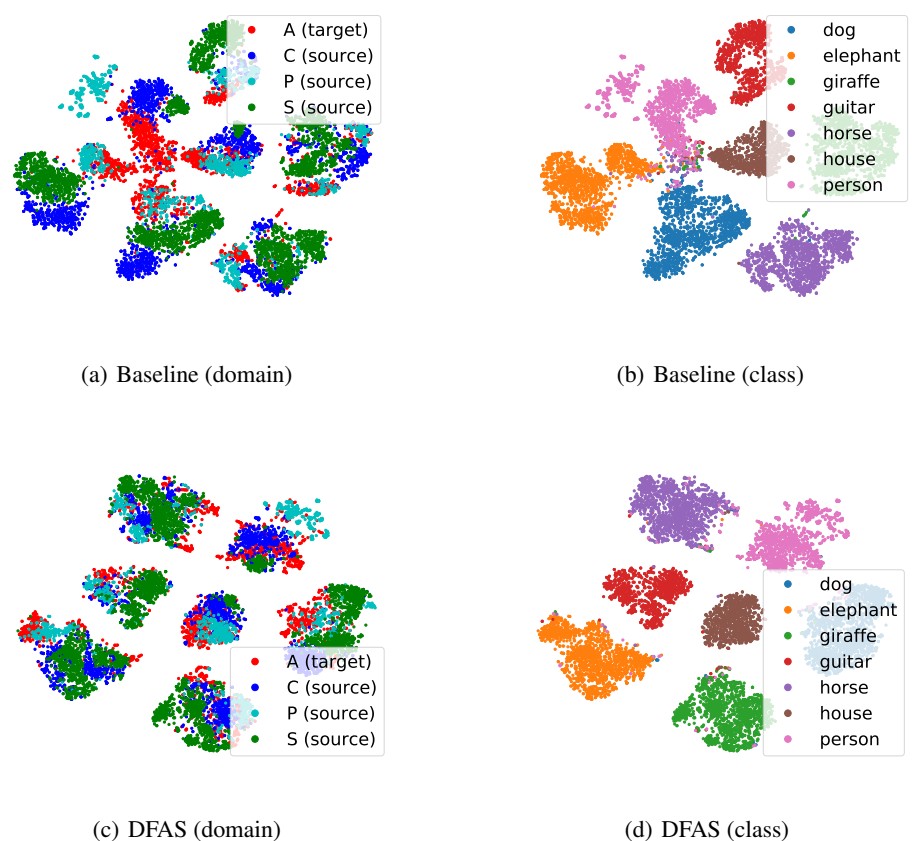

Figure 3: The t-SNE visualization of extracted features, using our proposed DFAS (a-b) and Baseline (c-d) on PACS dataset in *MSDS* setting. In (a) and (c), the different colors indicate different domains. In (b) and (d), the different colors indicate different classes.

## C  OPTIMIZATION ALGORITHM FOR FINDING THE HARDEST $S_v$

### C.1  OPTIMIZATION ALGORITHM

To solve the problem of

$$\max_{S_v, A} \sum_{(x,y) \in S_v} l\left(f_w(x), y\right) - \alpha \left\langle \nabla_w l(f_w(x), y), A \right\rangle \quad s.t. \ A = g_w^t, S_v \in \Gamma_\xi, \tag{11}$$

we design an alternative iteration algorithm in Sect. 3.2. Specifically, we initialize $A$ with gradient of a sample randomly selected from $S$. Then we alternatively update $S_v$ and $A$.

Given $A$, $S_v$ is updated by solving

$$\max_{S_v} \sum_{(x,y) \in S_v} l\left(f_w(x), y\right) - \alpha \left\langle \nabla_w l(f_w(x), y), A \right\rangle$$
$$s.t. \ S_v \subset S, |S_v| = \xi, \tag{12}$$

where the constraints are derived from the definition of $\Gamma_\xi$. Equation (12) indicates that the optimal $S_v$ consists of $\xi$ samples that have the largest values of $l\left(f_w(x), y\right) - \alpha \left\langle \nabla_w l(f_w(x), y), A \right\rangle$. Thus we compute and rank the values of $l\left(f_w(x), y\right) - \alpha \left\langle \nabla_w l(f_w(x), y), A \right\rangle$ for all $(x, y) \in S$ and select the largest $\xi$ samples to constitute the $S_v$.

Given $S_v$ ($S_t$ is then given), we update $A$ to satisfy the constraint $A = g_w^t$ in Eq. (11), then $A$ is

$$A = g_w^t = \frac{1}{|S_t|} \sum_{(x,y) \in S_t} \nabla_w l(f_w(x), y). \tag{13}$$

### C.2 CONVERGENCE IN EXPERIMENTS

We show empirically the convergence of this alternative iteration algorithm in Fig. 4, with the values of objective function in Eq. (11). Figure 4 shows that the values of objective function converges after only a few iterations.

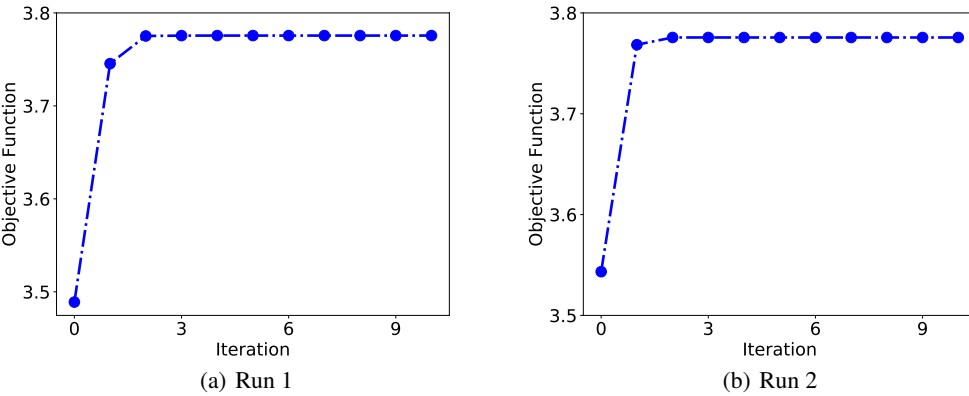

(a) Run 1           (b) Run 2

Figure 4: Convergence of the alternative iterations for finding the hardest $S_v$. (a) and (b) respectively show the values of objective function in Eq. (11) in two different runs with different initializations.

We also check if the splitting changes when the value of objective function converges. To do this, we count the ratio of changed sample indexes in $S_v$ at each iteration, as in Table 11. Table 11 shows that the splitting is not changed when the value of objective function converges.

Table 11: Ratio of changed sample indexes in $S_v$ at each iteration.

| Iteration | 1 | 2 | 3 | 4 | 5 | 6 |
|---|---|---|---|---|---|---|
| Ratio (%) | 44.0 | 4.5 | 0.0 | 0.0 | 0.0 | 0.0 |

### C.3 TOY EXAMPLE.

We present a toy example in this section to check if this algorithm can find the optimal solution. The toy example is an 2-dimensional classification problem, as shown in Fig. 5. Different colors indicate different classes. A fully-connected layer without bias is used as the network (learner). We split the data of the first class (blue points) with our algorithm. The candidate solutions with corresponding objective function values are given in Table 12.

Table 12: Candidate solutions (sample indexes) with objective function values of the toy example.

| Candidate $S_v$ | (0,1,2) | (0,1,3) | (0,1,4) | (0,1,5) | (0,2,3) | (0,2,4) | (0,2,5) |
|---|---|---|---|---|---|---|---|
| Objective function | 6.71 | 7.65 | 7.88 | 6.41 | 5.60 | 6.83 | 5.36 |
| Candidate $S_v$ | (0,3,4) | (0,3,5) | (0,4,5) | (1,2,3) | (1,2,4) | (1,2,5) | **(1,3,4)** |
| Objective function | 7.77 | 6.30 | 6.52 | 7.27 | 7.50 | 6.03 | **8.45** |
| Candidate $S_v$ | (1,3,5) | (1,4,5) | (2,3,4) | (2,3,5) | (2,4,5) | (3,4,5) | |
| Objective function | 6.98 | 7.20 | 7.39 | 5.93 | 6.15 | 7.10 | |

The solutions in the iteration process of our algorithm are reported in Table 13. The solutions converge to (1,3,4), which is the optimal solution in Table 12. This indicates that our algorithm can find the

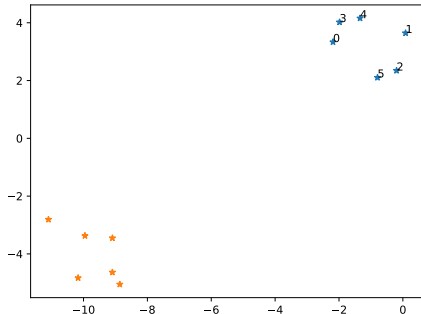

Figure 5: Toy data.

Table 13: Solutions of the iteration process.

| Iterations | 0 | 1 | 2 | 3 | 4 |
|---|---|---|---|---|---|
| Solution | (0,2,5) | (1.3,4) | (1,3,4) | (1,3,4) | (1,3,4) |
| Objective function | 6.89 | 8.43 | 8.45 | 8.45 | 8.45 |

optimal splitting for the toy example. We also give the code of this toy example as below, and the reader may rerun it to verify the results.

Code the toy example:

```python
import torch
import numpy as np
import matplotlib.pyplot as plt
from sklearn.datasets import make_blobs

#####
np.random.seed(1)
data = make_blobs(n_samples=12,centers=2)
x1 = data[0][data[1]==0]
x2 = data[0][data[1]==1]

plt.plot(x1[:,0],x1[:,1],'*')
plt.plot(x2[:,0],x2[:,1],'*')
t = 0
for x in x1:
  plt.text(x[0],x[1],str(t))
  t+=1
plt.savefig('samples.pdf')

###
model = torch.nn.Linear(2,2,bias=False).cuda()
torch.nn.init.xavier_uniform_(model.weight)
log = open('log.txt','w')
log.write('weight:'+ str(model.weight.cpu().data.numpy())+'\n')

feat = torch.Tensor(data[0]).cuda()
label = torch.LongTensor(data[1]).cuda()
loss_fun = torch.nn.CrossEntropyLoss()

Grads = []
Loss = []
for i in range(len(feat)):
  out =
      torch.nn.functional.linear(feat[i].view(1,-1),weight=model.weight,bias=None)
  loss_x = loss_fun(out,label[i].view(-1,))
```

```python
    grad = torch.autograd.grad(loss_x,model.weight)[0]
    Loss.append(loss_x.cpu().data.numpy())
    Grads.append(grad.cpu().data.numpy())

##we split the data of the first class
Loss = np.array(Loss)[data[1] == 0]
Grads = np.array(Grads)[data[1] == 0]
#
# np.save('toy_Grads.npy',Grads)
# np.save('toy_Loss.npy',Loss)

# Loss = np.load('toy_Loss.npy')[data[1] == 0]
# Grads = np.load('toy_Grads.npy')[data[1] == 0]
alpha = 0.001

def adv_loss(val_index=[0,1,2]):
    train_index = np.delete(np.arange(len(Loss)),val_index)
    loss = np.mean(Loss[val_index])
    grad_val = np.mean(Grads[val_index],axis=0)
    grad_train = np.mean(Grads[train_index],axis=0)

    return loss - alpha*np.sum(grad_val*grad_train)

#####brute force searching
Solutions = []
Values = []
#generate solutions
for i in range(len(Loss)-2):
    for j in range(i+1,len(Loss)-1):
        for k in range(j+1,len(Loss)):
            Solutions.append([i,j,k])
for val_idex in Solutions:
    Values.append(adv_loss(val_idex))

optimal_idx = np.array(Values).argmax()
optimal_solution = Solutions[optimal_idx]
optimal_value = max(Values)

print("all possible solutions:",Solutions)
print("objective function values of possible solutions:",Values)
print("optimal solution:",optimal_solution,"\nobjective function value
    of optimal solution:",optimal_value)

log.write(str({"all possible solutions":Solutions,
"objective function values of possible solutions":Values,
"optimal solution":optimal_solution,
"objective function value of optimal solution":optimal_value}))

###### our algorithm
A = Grads[np.random.randint(len(Loss))]
values_tmp = []
solutions_tmp = []
mtr_index =
    np.random.choice(np.arange(len(Loss)),size=len(Loss)//2,replace=False)
l = np.mean(Loss- alpha*np.sum(Grads*A.reshape((1,2,2)),axis=(1,2)))
values_tmp.append(l)
solutions_tmp.append(np.delete(np.arange(len(Loss)),mtr_index))
for i in range(5):
    D = np.sum(Grads*A,axis=(1,2))
    Loss_ = Loss-alpha*D
    idx_sort = np.argsort(Loss_)

    mtr_index = idx_sort[:len(Loss) // 2]
    mte_index = idx_sort[len(Loss) // 2 :]
```

```
    values_tmp.append(Loss_[mte_index].mean())
    solutions_tmp.append(mte_index)
    A = np.mean(Grads[mtr_index],axis=0)

print("our optimal solution:",solutions_tmp[-1],"\nobjective function
    value of our optimal solution:",values_tmp[-1])
log.write(str({"our optimal solution":solutions_tmp[-1],
"objective function value of our optimal solution":values_tmp[-1]}))
print(solutions_tmp,values_tmp)
```

## D    PROOF OF THEOREM 1

We first introduce VC-dimension based generalization bound and domain adaptation theory in Appendix D.1, then present two lemmas in Appendix D.2, and finally give the proof of Theorem 1 in Appendix D.3.

### D.1    PRELIMINARY

**VC-dimension based generalization bound.**

**Theorem A-1.** *(Abu-Mostafa et al., 2012) Let $S$ be the set of training data i.i.d. sampled for distribution $\mathcal{P}$. For any $\delta \in (0,1)$, with probability at least $1 - \delta$, we have $\forall h$ ($h : \mathcal{X} \to \{0,1\}$) in hypothesis space $\mathcal{H}$,*

$$|\epsilon_{\mathcal{P}}(h) - \hat{\epsilon}_S(h)| \leq \sqrt{\frac{8}{|S|}\left(VC(\mathcal{H})\log\frac{2e\,|S|}{VC(\mathcal{H})} + \frac{4}{\delta}\right)}. \tag{14}$$

*where $\epsilon_{\mathcal{P}}(h) = \mathbb{E}_{(x,y)\sim\mathcal{P}}[\mathbb{I}_{\{(h(x))\neq y\}}]$ and $\hat{\epsilon}_S(h) = \frac{1}{|S|}\sum_{(x,y)\in S}\mathbb{I}_{\{(h(x))\neq y\}}$.*

**Domain adaptation theory.**

**Theorem A-2.** *(Ben-David et al., 2007; 2010) For any $h$ in hypothesis space $\mathcal{H}$, we have*

$$\epsilon_{\mathcal{Q}}(h) \leq \epsilon_{\mathcal{P}}(h) + \frac{1}{2}d_{\mathcal{H}}(\mathcal{P},\mathcal{Q}) + \lambda^*, \tag{15}$$

*where $\lambda^* \geq \inf_{h'\in\mathcal{H}}\{\epsilon_{\mathcal{P}}(h') + \epsilon_{\mathcal{Q}}(h')\}$ and*

$$d_{\mathcal{H}}(\mathcal{P},\mathcal{Q}) = 2\sup_{h\in\mathcal{H}}|\mathbb{E}_{\mathcal{P}}[h=1] - \mathbb{E}_{\mathcal{Q}}[h=1]| \tag{16}$$

*is $\mathcal{H}$-divergence.*

### D.2    LEMMAS

**Lemma A-1.** *For any $S_v \in \Gamma_\xi$ and $S_t = S - S_v$, $\forall\delta \in (0,1)$, with probability at least $1 - \delta$, we have $\forall f \in \mathcal{H}_{S_t}$,*

$$|\epsilon_{\mathcal{P}}^{\Psi}(f) - \hat{\epsilon}_{S_v}^{\Psi}(f)| \leq \sqrt{\frac{8}{|S_v|}\left(VC(\mathcal{H}_{S_t}^{\Psi})\log\frac{2e\,|S_v|}{VC(\mathcal{H}_{S_t}^{\Psi})} + \frac{4}{\delta}\right)}, \tag{17}$$

*where $\epsilon_{\mathcal{P}}^{\Psi}(f) = \mathbb{E}_{(x,y)\sim\mathcal{P}}[\mathbb{I}_{\{\Psi(f(x))\neq y\}}]$ is generalization error on distribution $\mathcal{P}$, $\hat{\epsilon}_{S_v}^{\Psi}(f) = \frac{1}{|S_v|}\sum_{(x,y)\in S_v}\mathbb{I}_{\{\Psi(f(x))\neq y\}}$ is empirical error, $\mathcal{H}_{S_t}^{\Psi} = \{\Psi \circ f : f \in \mathcal{H}_{S_t}\}$, $VC(\mathcal{H}_{S_t}^{\Psi})$ is the VC-dimension of $\mathcal{H}_{S_t}^{\Psi}$, and $\Psi(\cdot)$ is the prediction rule such as the Bayes Optimal Predictor, i.e., $\Psi(f(x)) = \mathbb{I}_{\{f(x)\geq\frac{1}{2}\}}$.*

**Proof:**

From the definition of $\mathcal{H}_{S_t}^{\Psi}$, for any $f \in \mathcal{H}_{S_t}$, there exists a $h_f \in \mathcal{H}_{S_t}^{\Psi}$ such that $h_f = \Psi \circ f$. Applying Theorem A-1, with probability at least $1 - \delta$, we have $\forall f \in \mathcal{H}_{S_t}$,

$$
\begin{aligned}
|\epsilon_{\mathcal{P}}^{\Psi}(f) - \hat{\epsilon}_{S_v}^{\Psi}(f)| = & |\epsilon_{\mathcal{P}}(h_f) - \hat{\epsilon}_{S_v}(h_f)| \\
\leq & \sqrt{\frac{8}{|S_v|}\left(VC(\mathcal{H}_{S_t}^{\Psi})\log\frac{2e\,|S_v|}{VC(\mathcal{H}_{S_t}^{\Psi})} + \frac{4}{\delta}\right)}.
\end{aligned}
\tag{18}
$$

**Lemma A-2.** *For any $S_v \in \Gamma_{\xi}$ and $S_t = S - S_v$, let $\epsilon_{\mathcal{P}}^{\Psi}(g) = \inf_{f \in \mathcal{H}_{S_t}} \epsilon_{\mathcal{P}}^{\Psi}(f)$ and $\hat{\epsilon}_{S_v}^{\Psi}(h) = \inf_{f \in \mathcal{H}_{S_t}} \hat{\epsilon}_{S_v}^{\Psi}(f)$, then $\forall \delta \in (0, 1)$, with probability at least $1 - \delta$, we have*

$$
\epsilon_{\mathcal{P}}^{\Psi}(g) \geq \hat{\epsilon}_{S_v}^{\Psi}(h) - \sqrt{\frac{8}{|S_v|}\left(VC(\mathcal{H}_{S_t}^{\Psi})\log\frac{2e\,|S_v|}{VC(\mathcal{H}_{S_t}^{\Psi})} + \frac{4}{\delta}\right)}.
\tag{19}
$$

**Proof:**

From the definition of $g$ and $h$, we have $\hat{\epsilon}_{S_v}^{\Psi}(g) \geq \hat{\epsilon}_{S_v}^{\Psi}(h)$. $\forall \delta \in (0, 1)$, with probability at least $1 - \delta$, we have

$$
\begin{aligned}
\epsilon_{\mathcal{P}}^{\Psi}(g) - \hat{\epsilon}_{S_v}^{\Psi}(h) = & \epsilon_{\mathcal{P}}^{\Psi}(g) - \hat{\epsilon}_{S_v}^{\Psi}(g) + \hat{\epsilon}_{S_v}^{\Psi}(g) - \hat{\epsilon}_{S_v}^{\Psi}(h) \\
\geq & \epsilon_{\mathcal{P}}^{\Psi}(g) - \hat{\epsilon}_{S_v}^{\Psi}(g) \\
\geq & -\sqrt{\frac{8}{|S_v|}\left(VC(\mathcal{H}_{S_t}^{\Psi})\log\frac{2e\,|S_v|}{VC(\mathcal{H}_{S_t}^{\Psi})} + \frac{4}{\delta}\right)}.
\end{aligned}
\tag{20}
$$

Thus, Eq. (19) holds.

### D.3 Proof of Theorem 1

**Proof:**

We denote by $\mathcal{H}^{\Psi_l}$ the hypothesis space such that $\forall h \in \mathcal{H}^{\Psi_l}$,

$$
h(x) = \Psi_l(f(x)) = \begin{cases} 1 & \text{if } l(f(x), y) > \gamma, \\ 0 & \text{otherwise}, \end{cases}
\tag{21}
$$

for $f \in \mathcal{H}$. Then

$$
\begin{aligned}
d_{\mathcal{H}^{\Psi_l}}(\mathcal{P}, \mathcal{Q}) = & 2 \sup_{h \in \mathcal{H}^{\Psi_l}} \left| \mathbb{E}_{\mathcal{P}}[h = 1] - \mathbb{E}_{\mathcal{Q}}[h = 1] \right| \\
= & 2 \sup_{f \in \mathcal{H}} \left| \mathbb{E}_{\mathcal{P}}[\Psi_l(f(x)) = 1] - \mathbb{E}_{\mathcal{Q}}[\Psi_l(f(x)) = 1] \right| \\
= & 2 \sup_{f \in \mathcal{H}} \left| \mathbb{E}_{\mathcal{P}}[\mathbb{I}_{\{l(f(x),y)>\gamma\}}] - \mathbb{E}_{\mathcal{Q}}[\mathbb{I}_{\{l(f(x),y)>\gamma\}}] \right| \\
= & 2 \sup_{f \in \mathcal{H}} \left\{ \mathbb{E}_{\mathcal{Q}}[\mathbb{I}_{\{l(f(x),y)>\gamma\}}] - \mathbb{E}_{\mathcal{P}}[\mathbb{I}_{\{l(f(x),y)>\gamma\}}] \right\} \\
\leq & 2 \sup_{f \in \mathcal{H}} \mathbb{E}_{\mathcal{Q}}[\mathbb{I}_{\{l(f(x),y)>\gamma\}}] - 2 \inf_{f \in \mathcal{H}} \mathbb{E}_{\mathcal{P}}[\mathbb{I}_{\{l(f(x),y)>\gamma\}}].
\end{aligned}
\tag{22}
$$

In the fourth equation, we utilize the assumption that $\mathbb{E}_{\mathcal{Q}}[\mathbb{I}_{\{l(f(x),y)>\gamma\}}] \geq \mathbb{E}_{\mathcal{P}}[\mathbb{I}_{\{l(f(x),y)>\gamma\}}]$. Given any $S_v \in \Gamma_{\xi}$ and $S_t = S - S_v$, we replace $\mathcal{H}$ by $\mathcal{H}_{S_t}$, then

$$
\begin{aligned}
d_{\mathcal{H}_{S_t}^{\Psi_l}}(\mathcal{P}, \mathcal{Q}) \leq & 2 \sup_{f \in \mathcal{H}_{S_t}} \mathbb{E}_{\mathcal{Q}}[\mathbb{I}_{\{l(f(x),y)>\gamma\}}] - 2 \inf_{f \in \mathcal{H}_{S_t}} \mathbb{E}_{\mathcal{P}}[\mathbb{I}_{\{l(f(x),y)>\gamma\}}] \\
\leq & 2C_1 - 2 \inf_{f \in \mathcal{H}_{S_t}} \mathbb{E}_{\mathcal{P}}[\mathbb{I}_{\{l(f(x),y)>\gamma\}}]
\end{aligned}
\tag{23}
$$

where $C_1 = \sup_{S'_v \in \Gamma_\xi} \sup_{f \in \mathcal{H}_{S-S'_v}} \mathbb{E}_{\mathcal{Q}}[\mathbb{I}_{\{l(f(x),y)>\gamma\}}]$. Applying Theorem A-2, for any $f \in \mathcal{H}_{S_t}$, we have

$$\epsilon_{\mathcal{Q}}^{\Psi_l}(f) \leq \epsilon_{\mathcal{P}}^{\Psi_l}(f) + C_1 - \inf_{f' \in \mathcal{H}_{S_t}} \mathbb{E}_{\mathcal{P}}[\mathbb{I}_{\{l(f'(x),y)>\gamma\}}] + \lambda^*(S_v), \tag{24}$$

where $\lambda^*(S_v) \geq \inf_{f' \in \mathcal{H}_{S-S_v}}\{\epsilon_{\mathcal{P}}^{\Psi_l}(f') + \epsilon_{\mathcal{Q}}^{\Psi_l}(f')\}$. Let $C_3 = \sup_{S'_v \in \Gamma_\xi} \lambda^*(S'_v) \geq \sup_{S'_v \in \Gamma_\xi} \inf_{f' \in \mathcal{H}_{S-S'_v}}\{\epsilon_{\mathcal{P}}^{\Psi_l}(f') + \epsilon_{\mathcal{Q}}^{\Psi_l}(f')\}$, we have

$$\epsilon_{\mathcal{Q}}^{\Psi_l}(f) \leq \epsilon_{\mathcal{P}}^{\Psi_l}(f) + C_1 - \inf_{f' \in \mathcal{H}_{S_t}} \mathbb{E}_{\mathcal{P}}[\mathbb{I}_{\{l(f'(x),y)>\gamma\}}] + C_3. \tag{25}$$

Applying Lemma A-1 to the first term of right side in Eq. (25), $\forall \delta \in (0,1)$, with probability at least $1 - \delta$, we have $\forall f \in \mathcal{H}_{S_t}$,

$$\epsilon_{\mathcal{P}}^{\Psi_l}(f) \leq \hat{\epsilon}_{S_v}^{\Psi_l}(f) + \sqrt{\frac{8}{|S_v|}\left(VC(\mathcal{H}_{S_t}^{\Psi_l})\log\frac{2e|S_v|}{VC(\mathcal{H}_{S_t}^{\Psi_l})} + \frac{4}{\delta}\right)}. \tag{26}$$

Applying Lemma A-2 to the third term of right side in Eq. (25), $\forall \delta \in (0,1)$, with probability at least $1 - \delta$, we have

$$\inf_{f' \in \mathcal{H}_{S_t}} \mathbb{E}_{\mathcal{P}}[\mathbb{I}_{\{l(f'(x),y)>\gamma\}}] \geq \inf_{f' \in \mathcal{H}_{S_t}} \frac{1}{|S_v|} \sum_{(x,y) \in S_v} \mathbb{I}_{\{l(f'(x),y)>\gamma\}}$$
$$-\sqrt{\frac{8}{|S_v|}\left(VC(\mathcal{H}_{S_t}^{\Psi_l})\log\frac{2e|S_v|}{VC(\mathcal{H}_{S_t}^{\Psi_l})} + \frac{4}{\delta}\right)}. \tag{27}$$

Combining Eq. (25), (26), (27) and thanks to the union bound, for any $\delta \in (0,1)$, with probability at least $1 - 2\delta$, we have $\forall f \in \mathcal{H}_{S_t}$,

$$\epsilon_{\mathcal{Q}}^{\Psi_l}(f) \leq \hat{\epsilon}_{S_v}^{\Psi_l}(f) + B(S_v) + 2\sqrt{\frac{8}{|S_v|}\left(VC(\mathcal{H}_{S_t}^{\Psi_l})\log\frac{2e|S_v|}{VC(\mathcal{H}_{S_t}^{\Psi_l})} + \frac{4}{\delta}\right)} + C_3, \tag{28}$$

where $B(S_v) = C_1 - \inf_{f' \in \mathcal{H}_{S_t}} \frac{1}{|S_v|} \sum_{(x,y) \in S_v} \mathbb{I}_{\{l(f'(x),y)>\gamma\}}$. Using the fact that $|S_v| = \xi$ and let $C_2 = \sup_{S'_v \in \Gamma_\xi} VC(\mathcal{H}_{S-S'_v}^{\Psi_l})\log\frac{2e\xi}{VC(\mathcal{H}_{S-S'_v}^{\Psi_l})}$, we have

$$\epsilon_{\mathcal{Q}}^{\Psi_l}(f) \leq \hat{\epsilon}_{S_v}^{\Psi_l}(f) + B(S_v) + 2\sqrt{\frac{8}{|S_v|}\left(C_2 + \frac{4}{\delta}\right)} + C_3. \tag{29}$$

