# OpenReview forum: "Domain-Free Adversarial Splitting for Domain Generalization"
_ICLR.cc/2021/Conference — Reject_

### Official Review · AnonReviewer2 · 2020-10-26
**Missing key justification with major and minor issues**

**Rating:** 5
**Confidence:** 4

**Review:**

This paper proposes to unify adversarial training and meta-learning in domain-free generalization where labels of source domains are unavailable. To maximize the domain shift between the subsets of meta-train and meta-val, adversarial training is leveraged to find the worst-case train/val splits. Extensive experiments on benchmark datasets under different settings demonstrate the effectiveness of the proposed method.

Pros:
+ The idea of adversarial train/val splitting in meta-learning is interesting.

+ The paper provides extensive experiments on benchmark datasets under different settings with multiple/single source domains and achieves state-of-the-art results on both PACS and Office-Home datasets.

+ The paper provides an upper bound of the generalization error on unseen target domains, which is implicitly minimized by the proposed method.

Major Cons:
- The effectiveness of adversarial train/val splitting is not well justified. Typically, meta-learning based methods (MASF and MetaReg) have no overlap between domains used for meta-train and meta-val. This is how the domain discrepancy between meta-train and meta-val can be modeled. However, DFAS has the issue that the same domain may be used in both meta-train and meta-val, which significantly limits the domain transportation and may yield limited domain generalization capacity in comparison with MASF and MetaReg. So why DFAS outperforms MASF and MetaReg in terms of domain generalization? More experiments and discussion are suggested to justify this point.

- The difference between adversarial splitting and domain-label-based splitting is unclear.  Additional experiments are suggested to empirically show the difference.

- The proposed adversarial train/val splitting is computationally expensive since it needs to rank all samples in each iteration. According to the ablation results on PACS (Tab.5), improvements over randomly selected train/val splitting seem quite limited.
More details about the convergence of alternative iteration should be provided.

- Fig.4 shows the objective function remains unchanged after the 3rd iteration. Will the splitting results be changed after this stage?

Minor Cons:
- It would be better to see an ablation study on the effect of margin m in Eq.6.

- Standard deviation of the reported results should be provided.

After rebuttal:
It is highly appreciated that the authors provide additional evidence in response to my reviews. My previous concerns about effectiveness and efficiency are addressed to some extent, however, this also means the original submission needs significant modification to address the concerns. I like the idea in general and the problem is well-motivated but it needs more work for a complete version. I would encourage the authors to further improve the quality of the paper.

---

> ### Author Response · Authors · 2020-11-17
> **Response to Reviewer2 (2/2)**
>
> __R2-Q3__: Effectiveness of adversarial splitting over random splitting
>
> To show the effectiveness of adversarial splitting over random splitting, we conduct additional ablation experiments on PACS dataset in both MSDS and SSDS settings and report the results as below.
>
> Table R2-3: Ablation results on PACS in MSDS setting (Y: Yes, N: No)
> $L_2$-norm　Rand-split　Adv-split　　　　A　　C　　 P　　 S　　Avg
> 　　N　　　　N　　　　　N　　　　　80.2　75.5　95.9　70.1　80.4
> 　　Y　　　　 N　　　　　N　　　　　82.5　77.5　95.2　75.6　82.7
> 　　N　　　　 Y　　　　　N　　　　　80.6　76.8　95.6　78.1　82.8
> 　　N　　　　N　　　　　 Y　　　　　83.1　77.0　94.8　79.3　83.6
> 　　Y　　　 　 Y　　　　　N　　　　　83.2　78.6　95.8　79.5　84.3
> 　　Y 　　　　N　　　　　 Y　　　　　84.2　79.5　95.8　82.1　85.4
>
> Table R2-4: Ablation results on PACS in SSDS setting (Y: Yes, N: No)
> $L_2$-norm　Adv-split　Rand-split　A$\rightarrow$C　A$\rightarrow$P　A$\rightarrow$S　C$\rightarrow$A　C$\rightarrow$P　C$\rightarrow$S　P$\rightarrow$A　P$\rightarrow$C　P$\rightarrow$S　S$\rightarrow$A　S$\rightarrow$C　S$\rightarrow$P　 Avg
> 　　Ｎ　　　　N　　　　N　　　　63.7　95.6　　63.5　72.0　　86.5　73.3　　68.4　32.7　　42.2　41.6　　60.3　49.3　　62.4
> 　　 Y　　　　  N　　　　N　　　　64.5　95.4　　67.2　69.8　　87.2　73.3　　67.2　31.7　　38.5　44.3　　67.9　49.8　　63.1
> 　　Ｎ　　　　 Y　　　　N　　　　65.8　94.8　　64.4　71.2　　85.2　74.2　　68.7　32.2　　39.5　52.4　　64.6　56.6　　64.1
> 　　N　　　　 N　　　　 Y　　　　67.6　95.2　　66.2　76.9　　88.4　73.5　　70.4　30.5　　35.3　52.7　　67.7　57.3　　65.1
> 　　Y　　　　  Y　　　　 N　　　　 67.2　95.3　　63.7　70.8　　85.3　75.5　　70.0　37.2　　45.9　53.5　　64.5　55.8　　65.4
> 　　Y　　　　 N 　　　　Y　　 　　67.5　94.5　　67.1　69.0　　86.5　73.8　　70.2　36.1　　52.1　56.7　　67.9　57.4　　66.6
>
> In Table R2-3 and Table R2-4, $L_2$-norm denotes the $L_2 $-normalization (please see Sect. 3.3). The results show that, in Table R2-3, ($L_2$-norm + Adv-split) (85.4%) outperforms ($L_2$-norm + Rand-split) (84.3%) by 1.1%, and Adv-split (83.6%) outperforms Rand-split (82.8%) by 0.8% in MSDS setting. In Table R2-4, ($L_2$-norm + Adv-split) (66.6%) outperforms ($L_2$-norm + Rand-split) (65.4%) by 1.2%, and Adv-split (65.1%) outperforms Rand-split (64.1%) by 1.0% in SSDS setting. These results demonstrate that the adversarial splitting stably outperforms the random splitting in different experimental settings.
>
> __R2-Q4__: Computational cost of the adversarial splitting algorithm
>
> Since we only update the worst splitting per epoch, instead of at each step of updating parameters, the computational cost is only slightly higher than that of random splitting. To show this, we compare the total training times of the adversarial splitting and random spitting in the same number of steps (20000) as bellow.
>
> Table R2-5: Total training time (hour) of the adversarial splitting and random spitting
> 　　　　Adv-split　Rand-split
> Time　　6.23h　　　5.90h
>
> From R2-5, the training time of Adv-split is only 5.6% (0.33/5.90) higher than Rand-split.
>
> __R2-Q5__: Convergence of the alternative iteration algorithm
>
> The alternative iteration algorithm converges in our experiments. We have reported the target error curves in Appendix A to show this. Please see Fig. 2 for the details. We also find it converges on Office-Home and CIFAR-10 datasets. We will provide more training curves (e.g. losses) of more tasks in the revised version.
>
> __R2-Q6__: Will the splitting results be changed after the 3rd iteration?
>
> To show whether the splitting is changed after the 3rd iteration, we count the ratio of changed sample indexes in $S_v$ at each iteration, as in Table R2-6.
>
> Table R2-6: Ratio of changed sample indexes in $S_v$ at each iteration
> Iteration　　　1　　2　　3　　4　　5　　6
> Ratio (%)　　44.0　4.5　 0.0　 0.0　 0.0　0.0
>
> Table R2-6 shows that the splitting is not changed when the value of objective function converges.
>
> __R2-Q7__: Effect of margin $m$
>
> We analyze the effect of $m$ in MSDS setting on PACS dataset.
>
> Table R2-7: Analysis of hyper-parameter $m$ in task A on PACS in MSDS setting
> $m$　　　 0.10　　0.15　　0.20　　0.25　　0.30
> Acc (%)　83.9　　83.7　　84.2　　84.0　　83.6
>
> Table R2-7 shows that the result is not sensitive to the value of $m$.
>
> __R2-Q8__: Standard deviation
>
> We report standard deviations of the results of experiment on PACS in MSDS setting in Table R2-8.
>
> Table R2-8: Results with standard deviations of experiment on PACS in MSDS setting
> 　　　　　 A　　　　 C　　 　 　 P　　 　 　S 　　　Avg
> DFAS　84.2$\pm$0.1　79.5$\pm$0.3　95.8$\pm$0.1　82.1$\pm$0.4　85.4
>
> Table R2-8 shows that the results of DFAS are stable. Standard deviations of the results of other experiments will be included in the revised version.

---

> ### Author Response · Authors · 2020-11-17
> **Response to Reviewer2 (1/2)**
>
> Thanks for the comments and questions.
>
> __R2-Q1__: Why DFAS with adversarial splitting outperforms other meta-learning methods with domain-label-based splitting
>
> Due to variations of style, poses, sub-classes, etc., the internal inconsistency within dataset is complicated. Domain-label partially capture the inconsistency, while cannot cover all possible internal inconsistency. Our adversarial splitting method does not rely on the domain label. It iteratively finds the hardest train/val splitting to the learner to maximize the inconsistency and train the learner to generalize well for the hardest splitting, in an adversarial training way. This strategy more flexibly investigates the possible inconsistency within training dataset, adaptively to the learner, and can potentially enhance the generalization ability of learner.
>
> Table R2-1: Values of objective function in Eq. (5) of Adv-split and Label-split
> Learner　　　$w_1$　　$w_2$　　$w_3$　　$w_4$
> Adv-split　　 4.15　 4.28　 2.61　 1.28
> Label-split 　  3.72　 2.82　 1.29　 0.21
>
>
> In Table R2-1, we first empirically show that the domain-label-based splitting (denoted as Label-split) is not as hard as our adversarial splitting (Adv-split) to the learner. In Table R2-1, we report the values of objective function in Eq. (5) of Adv-split and Label-split by fixing the learner with different network parameters $w_i$ at different epoch (1th, 2th, 5th and 10th) in training process. Larger value in the table indicates that the splitting is harder to the learner (i.e., network). It can be observed that the domain-label-based splitting (Label-split) is not as hard as Adv-split to learner.
>
> We also conduct experiments on PACS in MSDS setting to fairly compare different splittings, including adversarial splitting (Adv-split), domain-label-based splitting (Label-split) and random splitting (Rand-split). The results are reported in Table R2-2.
>
> Table R2-2: Results of different splittings
> 　　　　　　　A　　C　　 P　　 S　　Avg
> Rand-split　  80.6　76.8　95.6　78.1　82.8
> Label-split　 81.2　75.9　94.7　80.1　83.0
> Adv-split　　83.1　77.0　94.8　79.3　83.6
>
> Table R2-2 shows that adversarial splitting outperforms random splitting and domain-label-based splitting. Note that we focus on domain-free setting, that we do not assume domain labels in training dataset. The domain-label-based splitting method cannot handle the setting that domain labels are unavailable.
>
> __R2-Q2__: Difference between adversarial splitting and domain-label-based splitting
>
> Domain-label-based splitting splits multiple domains into meta-train and meta-test domains, leveraging the domain labels. While in our adversarial splitting, the train/val subsets is specific to learner to enlarge the domain gap for the learner. Hence, our adversarial spitting is more flexible to capture possible internal inconsistency in training dataset. The results in Table R2-2 shows that when that training data are from multiple domains and domain labels are known, our adversarial splitting outperforms domain-label-based splitting. Moreover, when the training data are from only a single domain, our adversarial splitting also performs well (please see Table R2-4 in R2-Q3). However, domain-label-based splitting cannot be used in this setting, since there is no domain label.

---

### Official Review · AnonReviewer3 · 2020-10-28
**submission 784 review**

**Rating:** 6
**Confidence:** 2

**Review:**

The paper provides a novel way to combine meta-learning and adversarial training for domain generalisation. Different from existing methods, the authors propose to split the training dataset into train/val subsets in an iteratively adversarial way, regardless of domain labels, by which the model can be trained to learn to generalise well from training subset to val subset via meta-learning in each iteration.

Reason for score:

Overall, I vote for accepting. It’s ingenious that the paper proposes to ignore domain labels to enhance the learned model’s generalization ability towards domain shift problems. The major concern of mine is that the paper provides limited explanation of how the meta-learning algorithm, Model-Agnostic Meta-Learning (MAML), is utilized in the training process(see detailed in cons).

1. The paper proposes to discard domain labels when training a model to generalise well, by which the original training set containing several domains can be seen as a large labelless domain and then splitted into training/val subsets to simulate domain shift, hence MAML can be used here for the training process.

2. The authors incorporate the min-max optimization method following adversarial training to let the model be more effective in learning domain shifts between training/val subsets.

3. One of the paper’s contributions is that it surprisingly find that L2 -normalization can help mitigate gradient explosion in the MAML algorithm.

4. The paper provides comprehensive experiments of different domain shift settings as well as theoretical proofs to evaluate the proposed method, which are quite solid and convincing.

Cons:

1. It would be better if the paper provides more details in ablation study, for example, it mentions that DFAS-3 finds the hardest val-subset only based on loss by setting \alpha = 0 in Eqn. (5), there can be more analysis about the hyper-parameter of α .

---

> ### Author Response · Authors · 2020-11-17
> **Response to Reviewer3**
>
> Thanks for the positive comments on our paper.
>
> __R3-Q1__: More details of ablation study
>
> We add more ablation experiments on PACS dataset in both MSDS and SSDS settings, of which the results are reported in Table R3-1 and Table R3-2 respectively.
>
> Table R3-1: Ablation results on PACS in MSDS setting (Y: Yes, N: No)
> $L_2$-norm　Rand-split　Adv-split　　　　A　　C　　 P　　 S　　Avg
> 　　N　　　　N　　　　　N　　　　　80.2　75.5　95.9　70.1　80.4
> 　　Y　　　　 N　　　　　N　　　　　82.5　77.5　95.2　75.6　82.7
> 　　N　　　　 Y　　　　　N　　　　　80.6　76.8　95.6　78.1　82.8
> 　　N　　　　N　　　　　 Y　　　　　83.1　77.0　94.8　79.3　83.6
> 　　Y　　　　 Y　　　　　 N　　　　　83.2　78.6　95.8　79.5　84.3
> 　　Y 　　　　N　　　　　 Y　　　　　84.2　79.5　95.8　82.1　85.4
>
> Table R3-2: Ablation results on PACS in SSDS setting (Y: Yes, N: No)
> $L_2$-norm　Adv-split　Rand-split　A$\rightarrow$C　A$\rightarrow$P　A$\rightarrow$S　C$\rightarrow$A　C$\rightarrow$P　C$\rightarrow$S　P$\rightarrow$A　P$\rightarrow$C　P$\rightarrow$S　S$\rightarrow$A　S$\rightarrow$C　S$\rightarrow$P　 Avg
> 　　Ｎ　　　　N　　　　N　　　　63.7　95.6　　63.5　72.0　　86.5　73.3　　68.4　32.7　　42.2　41.6　　60.3　49.3　　62.4
> 　　 Y　　　　  N　　　　N　　　　64.5　95.4　　67.2　69.8　　87.2　73.3　　67.2　31.7　　38.5　44.3　　67.9　49.8　　63.1
> 　　Ｎ　　　　 Y　　　　N　　　　65.8　94.8　　64.4　71.2　　85.2　74.2　　68.7　32.2　　39.5　52.4　　64.6　56.6　　64.1
> 　　N　　　　 N　　　　 Y　　　　67.6　95.2　　66.2　76.9　　88.4　73.5　　70.4　30.5　　35.3　52.7　　67.7　57.3　　65.1
> 　　Y　　　　  Y　　　　 N　　　　 67.2　95.3　　63.7　70.8　　85.3　75.5　　70.0　37.2　　45.9　53.5　　64.5　55.8　　65.4
> 　　Y　　　　 N 　　　　Y　　 　　67.5　94.5　　67.1　69.0　　86.5　73.8　　70.2　36.1　　52.1　56.7　　67.9　57.4　　66.6
>
> In Table R3-1 and Table R3-2, $L_2$-norm denotes the $L_2 $-normalization (please see Sect. 3.3). Rand-split denotes that we randomly split the train/val subsets at each step of updating the parameters of network. Adv-split indicates that we update the worst splitting by solving the maximization problem in Eq. (5) per epoch. The results show that $L_2$-norm + Adv-split (i.e., DFAS) achieves the best performance in both MSDS (85.4%) and SSDS (66.6%) settings. $L_2$-norm and Adv-split are both effective in both MSDS and SSDS settings.
>
> __R3-Q2__: Analysis of hyper-parameter $\alpha$
>
> We evaluate the effect of $\alpha$ in MSDS setting on PACS dataset. The results are reported as below.
>
> Table R3-3: Analysis of hyper-parameter $\alpha$ in task A on PACS in MSDS setting
> $\alpha$　　　　1e-7　1e-6　1e-5　1e-4　1e-3
> Acc (%) 　83.４　83.9　84.2　83.7　82.6
>
> The ACC is stable to the values of $\alpha$ in large range of 1e-6 to 1e-4. Small $\alpha$ results in small step-size for parameter updating in meta-learning framework, and limits the benefits from meta-learning and adversarial splitting. Larger $\alpha$ results in larger step-size for gradient descent based network updates, which may fail to decrease the training loss from the optimization perspective.

---

### Official Review · AnonReviewer1 · 2020-10-29
**Interesting idea, but some points should be addressed**

**Rating:** 5
**Confidence:** 3

**Review:**

$Paper$ $summary$

This paper focuses on domain generalization, targeting the challenging scenario where the training set might not include different sources; even under the presence of different sources, the problem formulation does not takes into account domain labels. The proposed solution is based on meta-learning, following the path drawn by Li et al. AAAI 2018; the Authors propose to adversarially split the training set in meta-train and meta-validation sets, and then update the current model in a direction that fosters good generalization performance on the meta-test. Results on standard benchmarks are encouraging.

$Pros$

- The proposed idea is rather interesting, enabling to apply meta-learning solutions also in absence of domain labels. In particular, I like the idea of finding meta-train and meta-test splits in an adversarial fashion. This is crucial, since randomly splitting the training set in meta-train and meta-validation would not be helpful, since it would lead to episodes where meta-train and meta-test are iid.

- The Authors provide a theoretical interpretation of their approach (due to my background, I was not able to properly review it though)

$Cons$

- My main concern is related to the way the maximization problem is tackled, in the objective in Eq. (5). I have reviewed Appendix C, but I cannot understand how convergence would require so few iterations. Even restricting the size of $S_v$ as mentioned in Section 3., the number of possible sample combinations that generate couples ($S_v$, $S_t$) is huge -- if I understood the process correctly, then with $|S_v|=K$ and $|S|=N$, the number of combinations is $\binom{N}{K}$ -- and for each of them the gradients that lead to the meta-update are different. Could the authors comment on this? Am I missing something? Related to this point, I am also concerned by the ablation study in Table 5 - where it is shown that, while helpful, the adversarial strategy does not help very significantly in the whole picture.

- The overall writing could be improved. Sentences like "this model can be further transfored to a minimax problem" in the Abstract are not properly exposed, and there are several examples throughout the manuscript. There is a misconception related to prior work: Carlucci et al. 2019 (as well as Volpi et al. 2018) also tackles the case where the training data only comprises a single source domain. This should be clarified in the Introduction/Related work.

$Review$ $summary$

I like the idea this paper starts from, and I like the proposed solution. I still do not properly understand how the maximization problem at the core of the method is approached, and I believe that the paper needs some exhaustive proof checking to improve the overall writing. I look forward reading the Author response and iterating the discussion.

---- Post-rebuttal comments----

I thank the Authors for their explanations. Yet, I still believe that this work is not ready for publication. Random splitting and adversarial splitting perform very comparably (1% is not a lot), in my opinion casting some doubts on how meaningful the solution found to the proposed optimization problem is. The Authors included a toy example in the Appendix, but this did not mitigate my concerns on the original manuscript's experiments. I still believe that the core idea is very interesting, and hope that it will be further investigated by the Authors for a subsequent submission.

---

> ### Author Response · Authors · 2020-11-17
> **Response to Reviewer1 (2/2)**
>
> __R1-Q3__: Improving the overall writing
>
> Thanks for your suggestions. We will replace “This model can be further transformed to a min-max problem” by “To achieve this goal, we propose a min-max optimization problem” in Abstract. The similar sentences will be revised accordingly in Sect.1 and Sect. 3. We will carefully revise the whole paper to improve the overall writing.
>
> We will discuss the works of Carlucci et al. 2019 and Volpi et al. 2018 in the Introduction and Related Works. They tackle the setting that training set comprises a single domain, and the train and test data are from different domains. Our domain-free setting method is flexible without assumptions on the requirement of domain labels and different domains of training and test data. Moreover, in methodology, these two methods mainly focus on augmenting new data to increase training data. Our method adversarially splits training dataset to train learner via meta-learning approach to enhance the generalization ability of learner. Experimental results in Table 3 show that our proposed DFAS outperforms JiGen (Carlucci et al. 2019) (66.6% vs. 54.6%).

---

> > ### Comment · AnonReviewer1 · 2020-11-22
> > **Response to rebuttal**
> >
> > Many thanks for replying to my review.
> >
> > Regarding R1-Q1, I think that the problem the Authors are facing in this manuscript is inherently different than a standard clustering one (where k-means can be adopted). I do not think that theory behind k-means proves anything here. Indeed, there are two stages, a first one where the points are sub-divided, and second one where the gradients are actually computed. To my understanding, it is unreasonable to assume that an optimization problem like this would converge in a few iterations; I might naturally be wrong, but if this is the case then it needs to be proved (the current empirical results - see next point - do not back up the claim that the solver is converging to any meaningful solution).
> >
> > Regarding R1-Q2, I confirm my doubts on the improvements, since they seem quite marginal (always $\sim1$%). In theory, the Rand split should not bring any improvement at all, since in this case the meta-train and meta-test splits are (on average) iid. Also related to previous point, I think that the optimization problem should be more properly tackled, and that proceeding in this direction will bring stronger results. Can the Authors perhaps consider a very small, toy example where the brute force solution to the min-max split can be computed? Or at least, a very large number of candidate solutions. A comparison between the outcome solution from this experiments and the one found by the proposed algorithm would carry significant information.
> >
> > Regarding R1-Q3, Carlucci et al. 2019 tackle both cases (multi-source and single-source). I am not questioning the superiority of one approach to the other, but the sentence at the end of Section 2 "Moreover, our approach is flexible in handling the settings that the training data could be from a single or several domains" can be misleading, since it seems to suggest that previous work cannot do it (while, for example, Carlucci et al. 2019 can).
> >
> > I still believe that the idea is very interesting, but it requires some more work. I would strongly suggest the Authors to keep working on such min-max problem; I believe that the current solver is not getting close to a useful solutions (due to the marginal improvements with respect to Rand splits), but that the proposed approach would work significantly better otherwise.

---

> > > ### Author Response · Authors · 2020-11-24
> > > **Response to Reviewer1**
> > >
> > > Thanks for your comments.
> > >
> > > (1) To check if the alternative iteration algorithm for solving the maximization problem can find the optimal solution, we have added a toy example in Appendix C.3. In this toy example, we first list all possible solutions and find the optimal one that achieves the maximum value of objective function. Then we run our alternative iteration algorithm to return a solution. It is shown that the returned solution is the optimal solution. Since it is non-trivial to analyze the convergence speed theoretically, we’ll keep working on it in our future works.
> > >
> > > (2) About the effectiveness of random splitting (Rand-split), we believe it is reasonable that random splitting is effective over Baseline for the following reason. Though meta-train and meta-test data are i.i.d. on average, there exists domain gap between meta-train samples and meta-test samples at each step because both meta-train and meta-test samples partially reflect the whole data. Hence, training the network with meta-learning approach forces the network, updated with meta-train samples, to improve its performance on the different meta-test samples. This may improve the generalization ability of the trained network when there exists domain gap. Since adversarial splitting (Adv-split) uniformly outperforms random splitting by around 1% in different settings and costs only slightly higher computational time than random splitting (please see R2-Q4 in the response to Reviewer2), we believe it is a useful technique for domain generalization.
> > >
> > > (3) We have removed the sentence “Moreover, our approach is flexible in handling the settings that the training data could be from a single or several domains”. Similar sentences have also been removed.

---

> ### Author Response · Authors · 2020-11-17
> **Response to Reviewer1 (1/2)**
>
> Thanks for the positive comment that “the proposed idea is rather interesting”. We clarify the raised questions as follows.
>
> __R1-Q1__: How would convergence require so few iterations in the maximization problem in Eq. (5)
>
> Thanks for this question. The optimization problem in Eq. (5) is similar to a 1 dimensional 2-cluster k-means algorithm, where $A$ corresponds to the center of the k-means algorithm and $S_v$ corresponds to the grouped samples. Similar to k-means algorithm, the optimization problem in Eq. (5) is a non-convex combinatorial problem. Due to its non-convexity, starting from a random initialization, our algorithm can quickly find a local minimizer. Since it resembles k-means algorithm, let us recall the k-means algorithm, which also solves a combinatorial non-convex problem. (Tang & Monteleoni, 2017) proves that k-means enjoys linear convergence. (Bottou & Bengio, 1995) showed in experiments that k-means converges after a few (around three) iterations. Our algorithm, also converges after a few iterations, similar to k-means algorithm. We are interested in further theoretically analyzing the convergence rate of our algorithm in future work.
>
> [1] Bottou, Leon, and Yoshua Bengio. "Convergence properties of the k-means algorithms." Advances in neural information processing systems (NeurIPS). 1995.
> [2] Tang, Cheng, and Claire Monteleoni. "Convergence rate of stochastic k-means." Artificial Intelligence and Statistics. PMLR, 2017.
>
> __R1-Q2__: Effectiveness of adversarial splitting strategy
>
> To illustrate the effectiveness of adversarial splitting (Adv-split) over random splitting (Rand-split), we compare the performance of learner trained with these two splitting strategies, as in Table 5 and the following additional ablation results on PACS dataset in both MSDS and SSDS settings.
>
> Table R1-1: Ablation results on PACS in MSDS setting (Y: Yes, N: No)
> $L_2$-norm　Rand-split　Adv-split　　　　A　　C　　 P　　 S　　Avg
> 　　N　　　　N　　　　　N　　　　　80.2　75.5　95.9　70.1　80.4
> 　　Y　　　　 N　　　　　N　　　　　82.5　77.5　95.2　75.6　82.7
> 　　N　　　　 Y　　　　　N　　　　　80.6　76.8　95.6　78.1　82.8
> 　　N　　　　N　　　　　 Y　　　　　83.1　77.0　94.8　79.3　83.6
> 　　Y　　　　 Y　　　　　 N　　　　　83.2　78.6　95.8　79.5　84.3
> 　　Y 　　　　N　　　　　 Y　　　　　84.2　79.5　95.8　82.1　85.4
>
> Table R1-2: Ablation results on PACS in SSDS setting (Y: Yes, N: No)
> $L_2$-norm　Adv-split　Rand-split　A$\rightarrow$C　A$\rightarrow$P　A$\rightarrow$S　C$\rightarrow$A　C$\rightarrow$P　C$\rightarrow$S　P$\rightarrow$A　P$\rightarrow$C　P$\rightarrow$S　S$\rightarrow$A　S$\rightarrow$C　S$\rightarrow$P　 Avg
> 　　Ｎ　　　　N　　　　N　　　　63.7　95.6　　63.5　72.0　　86.5　73.3　　68.4　32.7　　42.2　41.6　　60.3　49.3　　62.4
> 　　 Y　　　　  N　　　　N　　　　64.5　95.4　　67.2　69.8　　87.2　73.3　　67.2　31.7　　38.5　44.3　　67.9　49.8　　63.1
> 　　Ｎ　　　　 Y　　　　N　　　　65.8　94.8　　64.4　71.2　　85.2　74.2　　68.7　32.2　　39.5　52.4　　64.6　56.6　　64.1
> 　　N　　　　 N　　　　 Y　　　　67.6　95.2　　66.2　76.9　　88.4　73.5　　70.4　30.5　　35.3　52.7　　67.7　57.3　　65.1
> 　　Y　　　　  Y　　　　 N　　　　 67.2　95.3　　63.7　70.8　　85.3　75.5　　70.0　37.2　　45.9　53.5　　64.5　55.8　　65.4
> 　　Y　　　　 N 　　　　Y　　 　　67.5　94.5　　67.1　69.0　　86.5　73.8　　70.2　36.1　　52.1　56.7　　67.9　57.4　　66.6
>
> Rand-split: In random splitting strategy, we randomly split the train/val subsets at each step of updating the parameters of learner.
>
> Adv-split: In adversarial splitting model, we update the worst splitting by solving the maximization problem in Eq. (5) per epoch. Compared with random splitting, adversarial splitting will find the hardest splitting to the learner. Hence, training with the hard train/val splitting may enhance the robustness of learner. This is verified by the results in Table R1-1 and Table R1-2.
>
> In Table R1-1 and Table R1-2, $L_2$-norm denotes the $L_2$-normalization. In Table R1-1, ($L_2$-norm + Adv-split) (85.4%) outperforms ($L_2$-norm + Rand-split) (84.3%) by 1.1% and Adv-split (83.6%) outperforms Rand-split (82.8%) by 0.8% in MSDS setting. In Table R1-2, ($L_2$-norm + Adv-split) (66.6%) outperforms ($L_2$-norm + Rand-split) (65.4%) by 1.2% and Adv-split (65.1%) outperforms Rand-split (64.1%) by 1.0% in SSDS setting. These results demonstrate that the adversarial splitting model outperforms the random splitting strategy in different experimental settings.

---

### Official Review · AnonReviewer4 · 2020-10-30
**Review for the paper**

**Rating:** 5
**Confidence:** 4

**Review:**

##########################################################################
Summary:

The paper proposes a new approach for domain generalization which minimizes the generalization error across train-validation split with the largest domain gap. The paper further gives theoretical bound on the generalization error of the proposed method.

##########################################################################
Pros:

The paper proposes a new approach for domain generalization. The idea of perform meta-learning from the train to validation split is already employed by previous works (Balaji et al., 2018; Li et al., 2019b; 2018a; Dou et al., 2019). The novelty of the paper falls mostly on performing meta-learning on the train-validation split with the largest domain gap.

The paper gives a theoretical bound on the generalization error to unseen target domain.

##########################################################################
Cons:

The paper needs to update the model parameter, the initialized parameters and the train-validation split, which may not converge or converges very slowly. Though empirical results show that the convergence of the method is fast, some theoretical demonstrations are needed for the convergence speed of different updates.

The contribution of the paper is a little incremental. Meta-learning based domain generalization methods are not novel. The only novelty of the paper is performing meta-learning on the train-validation split with the largest domain gap, which is an incremental contribution over meta-learning based domain generalization (demonstrated by the ablation study).

As shown in Table 1, Baseline w/L2 outperforms Baseline by 2.3% but DFAS outperforms Baseline w/L2 only by 2.7%. This means that the main performance gain is from L2 while the other contributions has very small performance gain (less than 1%). However, L2-normalization (Finn et al., 2017; Dou et al., 2019) is a widely-used techniques for meta-learning and domain generalization, which is not counted as a novel contribution for the paper. The authors need to further demonstrate that the main contribution: meta-learning on the train-validation split with the largest domain gap, has huge performance gain.

##########################################################################
I lean to rejecting the paper since the performance gain is mostly falls on the L2-normalization but the main contribution of the paper: performing meta-learning across the train-val split with the largest domain gap, does not have much performance gain.

---

> ### Author Response · Authors · 2020-11-17
> **Response to Reviewer4 (2/2)**
>
> __R4-Q3__: Clarifying ablation results of the performance contributions of adversarial splitting and feature $L_2$-normalization
>
> We respectfully disagree on the comment that “Baseline w/$L_2$ outperforms Baseline by 2.3% but DFAS outperforms Baseline w/$L_2$ only by 2.7%. This means that the main performance gain is from $L_2$ while the other contributions has very small performance gain (less than 1%)” .
>
> We first recall the definitions of different methods in Table 5 (not Table 1).
> Baseline: Aggregating three source domains to train learner.
> Baseline w/$L_2$: Applying feature $L_2$-normalization over Baseline.
> DFAS: Performing both adversarial splitting (Eq. (2)) and feature $L_2$-normalization over Baseline.
> DFAS-1: Only performing adversarial splitting model (Eq. (2)) without utilizing feature $L_2$-normalization over Baseline.
>
> The results in Table 5 shows that Baseline w/$L_2$ (82.7%) outperforms Baseline (80.4%) by 2.3%. DFAS (85.4%) outperforms Baseline (80.4%) and Baseline w/$L_2$ (82.7%) by 5.0% and 2.7% respectively (i.e., 5.0% = 2.7% + 2.3%). This indicates that both adversarial splitting (contributed 2.7% improvement) and feature $L_2$-normalization (contributed 2.3% improvement) significantly contribute to total performance gain of 5.0% over Baseline. Moreover, DFAS-1 (only with additional adversarial splitting over Baseline) (83.6%) outperforms Baseline (80.4%) by 3.2%, indicating that the adversarial splitting is indeed effective.
>
> __R4-Q4__: Theoretical demonstrations for convergence speed
>
> Thanks for this question. It is an interesting research direction to analyze convergence speed of our adversarial training algorithm. Our min-max optimization is based on iteratively updating the worst-case splitting and the parameters of learner with meta-learning. As a deep meta-learning-based training algorithm, both these two sub-optimization problems are non-convex problems. It is interesting but non-trivial to analyze their convergence speed. We have reported the target error curves in Fig. 2 in Appendix A, and empirically find that our algorithm converges in our training tasks.
>
> In the adversarial training algorithm, we only update the worst-case splitting per epoch. The training time of our method is only slightly longer than that of random-splitting-based meta-learning method (6.23 hours vs. 5.90 hours). Please refer to R2-Q4 in the response to Reviewer2 for the discussion on computational time.
>
> As this work mainly focuses on the novel adversarial training framework with theoretical analysis on generalization error to target domain from the machine-learning theory perspective, its convergence speed analysis from the optimization perspective will be left in our future work.

---

> ### Author Response · Authors · 2020-11-17
> **Response to Reviewer4 (1/2)**
>
> In the followings, we will clarify the concerns on novelty, effectiveness of adversarial splitting, convergence speed, etc. We will also carefully revise paper according to these questions.
>
> __R4-Q1__: Clarifying the contributions
>
> To the best of our knowledge, this is the first work that models domain splitting and meta-learning in a single min-max adversarial training problem. It improves the domain generalization ability of learned network by adversarial splitting of train/val domains to challenge the network to improve its generalization to large domain gaps. This idea is also theoretically consistent with our theoretical findings on domain generalization in Sect. 4, which is also our novel contribution.
>
> Because of the possible misunderstanding on our results in Table 5, we respectfully disagree on the comments that “performance gain is from $L_2$-normalization while the other contributions has very small performance gain”. Please see R4-Q3 for the quantitative results on the performance contribution of adversarial splitting. Our adversarial splitting method indeed significantly improves the domain generalization results.
>
> Our contributions are listed as follows.
> (1)	We consider the more general domain generalization (DG) setting that we do not assume that there exist different domains in training dataset and do not assume that the domain labels are available.
> (2)	We propose a novel adversarial network training framework for DG, by unifying adversarial domain splitting and meta-learning in a principled min-max optimization problem.
> (3)	We also present an upper bound of the generalization error on unseen target domain. The terms in this upper bound are implicitly minimized by our method.
> (4)	We find that feature $L_2$-normalization is effective and can mitigate gradient explosion for DG. The underlying reason is theoretically analyzed.
> (5)	Extensive experiments under three different settings (please refer to Sect. 5) show SOTA results achieved by our approach.
>
>
> __R4-Q2__: Novelty of feature $L_2$-normalization in domain generalization
>
> Feature $L_2$-normalization has been wildly used in face recognition (Liu et al., 2017; Wang et al., 2018) and domain adaptation (Saito et al., 2019; Gu et al., 2020). However, to the best of our knowledge, there are rare work that apply $L_2$-normalization to domain generalization. The suggested papers (Finn et al., 2017; Dou et al., 2019) utilize gradient-clipping rather than feature $L_2$-normalization to mitigate gradient explosion.
>
> In the context of domain generalization, we find that feature $L_2$-normalization not only improves the performance of learner (please see Table 5 in Sect. 5.4), but also mitigates gradient explosion (please see Appendix A) that occurs frequently during the training of meta-learning for DG. We also theoretically analyze the underlying reason, and present Proposition 1 in Sect. 3.3, for analyzing the gradient norm in the training process of meta-learning for DG.
>
> We will carefully revise the claims on the novelty and our theoretical analysis on $L_2$-normalization in DG, considering these related works.

---

### Decision · Program_Chairs · 2021-01-07
**Final Decision**

**Decision:**

Reject

**Comment:**

The paper is proposing a domain generalization method based on the intuition that an invariant model would work for any split of train/val. Hence, the method uses adversarial train/val splits during training. The paper is reviewed by three expert reviews and none of them championed the paper to be accepted. I carefully checked the reviews and the authors' response and agree with the reviewers. Specifically:

- R#1: Argues that the paper is not ready for publication. Also argues the optimization problem is only a motivation as it is not directly solved. This is an important issue and it needs to be addressed in a conclusive manner.
- R#2: Argues empirical studies do not show the value of train/val splitting. I partially disagree with this issue but it is clear that more qualitative and quantitative study is needed to properly justify the proposed method.
- R#3: Argues the contribution is not enough for publication. The paper is clearly novel but the contribution and novelty is not presented in a clear manner. Moreover, the empirical study does not complement the novelty. Hence, I disagree with the comment.

Overall, I believe the paper proposes an interesting idea. However, the presentation and empirical studies need to be improved significantly. I recommend authors to address these issues and submit to the next conference.